# The Effects of Cyanobacterial Bloom Extracts on the Biomass, Chl-a, MC and Other Oligopeptides Contents in a Natural *Planktothrix agardhii* Population

**DOI:** 10.3390/ijerph17082881

**Published:** 2020-04-22

**Authors:** Magdalena Toporowska, Hanna Mazur-Marzec, Barbara Pawlik-Skowrońska

**Affiliations:** 1Department of Hydrobiology and Protection of Ecosystems, University of Life Sciences in Lublin, Dobrzańskiego 37, 20-262 Lublin, Poland; barbara.pawlik@up.lublin.pl; 2Division of Marine Biotechnology, University of Gdańsk, Al. Marszałka Piłsudskiego 46, 81-378 Gdynia, Poland; hanna.mazur-marzec@ug.edu.pl

**Keywords:** cyanobacteria, cyanotoxins, microcystins, aeruginosins, anabaenopeptins, cyanopeptolins, non-ribosomal peptides, chemotypes

## Abstract

Blooms of the cyanobacterium *Planktothrix agardhii* are common in shallow, eutrophic freshwaters. *P. agardhii* may produce hepatotoxic microcystins (MCs) and many other bioactive secondary metabolites belonging mostly to non-ribosomal oligopeptides. The aim of this work was to study the effects of two extracts (Pa-A and Pa-B) of *P. agardhii*-predominated bloom samples with different oligopeptide profiles and high concentration of biogenic compounds on another natural *P. agardhii* population. We hypothesised that the *P. agardhii* biomass and content of oligopeptides in *P. agardhii* is shaped in a different manner by diverse mixtures of metabolites of different *P. agardhii*-dominated cyanobacterial assemblages. For this purpose, the biomass, chlorophyll a and oligopeptides content in the treated *P. agardhii* were measured. Seven-day microcosm experiments with four concentrations of the extracts Pa-A and Pa-B were carried out. Generally, aeruginosins (AERs), cyanopeptolins (CPs) and anabaenopeptins (APs) were the most numerous peptides; however, only 16% of them were common for both extracts. The addition of the extracts resulted in similar effects on *P. agardhii*: an increase in biomass, Chl-a and MC content in the exposed *P. agardhii* as well as changes in its oligopeptide profile were observed. MCs present in the extracts did not inhibit accumulation of *P. agardhii* biomass, and did not have any negative effect on MC and Chl-a content. No evidence for bioaccumulation of dissolved peptides in the *P. agardhii* exposed was found. As the two tested extracts differed considerably in oligopeptide composition, but contained similar high concentrations of nutrients, it seems that biogenic compounds, not oligopeptides themselves, positively influenced the mixed natural *P. agardhii* population.

## 1. Introduction

In recent decades, global warming and increasing water eutrophication have intensified harmful cyanobacterial blooms (‘cyanoHABs’) formed by different species of planktonic cyanobacteria [1,2,3]. Cyanobacterial blooms can harm humans and affect the functioning of aquatic biocenoses [2,4,5], particularly when they are composed of strains producing cyanotoxins such as hepatotoxic microcystins and cylindrospermopsins, neurotoxic anatoxin-a, anatoxin-a(s), saxitoxins or other less known compounds [6,7,8,9]. Filamentous *Planktothrix agardhii* is one of the most common potentially toxigenic bloom-forming cyanobacterium occurring in shallow, eutrophic freshwaters of temperate climatic zones [10,11,12]. Ongoing worldwide eutrophication problems [13] may intensify water blooms caused by this species. For example, Doculil and Teubner [14] showed that the biomass of *P. agardhii* increased with increasing concentrations of total phosphorus. In recent years, the bioactivity and role of metabolites produced by *P. agardhii* warrant more and more attention [12,15,16,17,18]. *P. agardhii* can produce toxic microcystins (MCs), and other, non-ribosomal oligopeptides (NRPs), such as aeruginosins (AERs), anabaenopeptins (APs), cyanopeptolins (CPs) and microginins (MRGs), as well as ribosomal microviridins (MICs) [7,10]. However, the development of particular toxic and/or non-toxic subpopulations within cyanobacteria community is impossible to predict [19]. Additionally, among *P. agardhii* populations, both MC-producing and non-MC-producing strains occur, and the genotypic variation concerns both MCs and other biologically active oligopeptide synthesis genes [12]. Therefore, the populations of *P. agardhii* are not clonal and may be composed of several chemotypes, characterised by different peptide patterns [17]. According to Rohrlack et al. [15] and Agha and Quesada [20], the peptide profile in a chemotype is basically constant; however, environmental conditions may slightly change production of specific peptides.

Cyanobacterial NRPs (including MCs) occur mainly intracellularly; however, during and after the bloom decay, they may be found in high concentrations in lake water. For example, Nasri et al. [21] reported up to 712 µg/L of dissolved MCs during natural *Microcystis* bloom collapse, whereas Jones and Orr [22] reported up to 1800 μg of MCs/Lafter treatment of bloom with an algicide. Peptides comprise more than 60% of the known biologically active compounds produced by cyanobacteria [23]. In addition to more than 240 MC isoforms, a minimum of 500 oligopeptides ranging from 400–1900 Da have been structurally identified to date [24] and references therein. While current research is strongly focused on exploring new oligopeptide variants and their bioactive properties, the biological role of these compounds remains elusive [8,20]. A number of biological functions of oligopeptides have been proposed, including roles as defence compounds against grazers [4,25], allelopathic metabolites active against eukaryotic algae [26], phytotoxic substances [18,27] or antiparasitic chemicals [16]. A few studies have reported the possible role of oligopeptides as info-chemicals involved in quorum sensing [28,29] or compounds involved in bloom formation and breakdown [12,30,31,32]. For example, a high frequency of genes involved in the synthesis of several toxins and other bioactive peptides were detected in bloom-forming cyanobacteria *P. agardhii* and *P. rubescens*, but not in other non-bloom-forming *Planktothrix* species [12]. This fact suggests a functional linkage between bioactive oligopeptide production and the potential for colonisation, niche construction and dominance in freshwater bodies [12] and references therein. Nevertheless, none of the hypotheses mentioned above have been evidenced and the role of cyanobacterial oligopeptides remains obscure and is still a matter of intense scientific discussion. In addition to peptides, which may be present in large numbers and quantities in cyanobacterial biomass [17,18,21,24], the presence of other components, such as biogenic compounds or minerals, may influence phytoplankton communities. For example, Suikkanen et al. [33] showed that cyanobacterial filtrates containing unknown compounds stimulated both colonial (*Snowella* spp.) and filamentous cyanobacteria (*Pseudanabaena* spp., *Anabaena* spp., *Aphanizomenon* sp., *N. spumigena*), a chlorophyte (*Oocystis* sp.), a dinoflagellate (*Amphidinium* sp.) and nanoflagellates, but inhibited cryptophytes. Further studies on the biological activity and role of oligopeptides and other cyanobacterial compounds in the development and toxicity of bloom-forming cyanobacteria are definitely required. It is also important that in nature, the majority of cyanobacterial blooms are multispecies, and many other microorganisms, such as some bacteria, viruses, fungi and protozoa, may play a direct or indirect role in cyanobacterial development, lysis events and transformation of molecules released from cyanobacterial cells [32,34]. Therefore, both interspecific and intraspecific interactions may occur.

The aim of this work was to study the effects of the extracts of two *P. agardhii*-predominated bloom samples characterised by different oligopeptide profiles and high concentration of dissolved and total nitrogen and phosphorus on another natural *P. agardhii* population. We hypothesised that the *P. agardhii* biomass and content of oligopeptides in *P. agardhii* is shaped in a different manner by diverse mixtures of metabolites of different *P. agardhii*-dominated cyanobacterial assemblages.

## 2. Materials and Methods

### 2.1. Sampling

A dense cyanobacterial scum sample predominated by *P. agardhii* (95% of the total cyanobacterial biomass) was collected in autumn 2010 from the littoral zone of the small (8 ha) and shallow (max. depth 3.2 m) hypertrophic Lake Syczyńskie (Eastern Poland; 51°17′12″ N, 23°14′16″ E) (Table 1, Figure 1). The second dense scum sample predominated (82%) by *P. agardhii* was collected in autumn 2013 in the littoral zone of the larger (38 ha) and deeper (max. depth 15.6 m) eutrophic Lake Czarne Sosnowickie (Czarne S.; 51°30′55″42 N, 23°01′37″96 E; Eastern Poland). After collection and phytoplankton enumeration (described in Section 2.3), the scum samples were frozen (at −20 °C) until the day of extraction, analyses and experiments. The scum samples were used to prepare the extracts Pa-A and Pa-B (respectively) that were used in experiments. The extraction procedure is described in Section 2.4.

Integrated water sample (3 L) containing natural phytoplankton community predominated (96%) by *P. agardhii* was collected with the aid of a hosepipe sampler in July 2015 from the surface to 0.5 m above the bottom in the central part of Lake Syczyńskie (Table 1, Figure 1). Simultaneously, subsamples of 100 mL and 500 mL were collected for the analysis of phytoplankton, microcystins and chlorophyll-a, respectively.

### 2.2. Experiments

The experimental design and conditions are presented in Figure 1. The natural phytoplankton community, predominated by *P. agardhii* (96% of the total phytoplankton biomass), present in crude lake water collected from Lake Syczyńskie was exposed (the same day) to the crude cyanobacterial extracts Pa-A and Pa-B obtained from *P. agardhii*-dominated scum samples collected in different years and lakes (Table 1, Figure 1). At the beginning of the experiments, the composition and abundance of *P. aghardii* population in each experimental condition was uniform as we used one and the same water sample that was mixed up for homogeneity before setting up experiments. The experiments were performed for seven days. Four dilutions of the extracts were used. The initial concentration range of extracellular Chl-a (an equivalent of cyanobacterial biomass in water) was similar in both experiments, i.e., 0.33–2.58 mg/L in the treatments with the extract Pa-A and 0.33–2.63 mg/L in the treatments with the extract Pa-B were used (Figure 1). The initial concentration of dissolved MCs in treatments with extracts Pa-A and Pa-B ranged from 4 to 32 µg/L and from 53 to 410 µg/L, respectively (Figure 1). The MC concentrations were environmentally relevant [22]. The concentrations of Chla, MCs and oligopeptides in the extracts were determined before use in the experiments. The experiments were carried out in 100 mL cups in a cabinet with a light: dark cycle (15:9 h), light intensity of 45 µmol/m^2^/s and at 21 °C. Each experimental variant (of 40 mL volume) including controls was set in triplicate. The initial biomass of *P. agardhii* in all experimental conditions (including controls) was 105 mg/L (Table 1), corresponding to cyanobacterial bloom conditions (an exponential phase) in lakes. The initial content of Chl-a in *P. agardhii* biomass was 3.4 µg/mg, whereas the content of extracellular MCs was 362 µg/ mg (Table 1). The cups containing *P. agardhii* control and *P. agardhii* exposed to the extracts were gently mixed once a day.

After exposure, the following analyses were carried out: phytoplankton enumeration, determination of phytoplankton biovolume, determination of cyanobacteria species contribution to the total phytoplankton biomass, content of Chl-a and MCs in *P. agardhii* biomass, and the number and relative content of oligopeptides other than MCs in the *P. agardhii* control and in the *P. agardhii* exposed to the highest concentration of the extracts. Dissolved MCs were analysed in filtered (GF/C Whatman filters) water obtained from the variants with the highest concentrations of the extracts to control their stability during the experiment.

### 2.3. Phytoplankton Enumeration, Chlorophyll-a, Nitrogen and Phosphorus Analyses

After collection and before freezing (at −20 °C), both cyanobacterial scum samples were analysed for taxonomic composition and cyanobacterial enumeration in a plankton Sedgewick-Rafter chamber under a light microscope. Cyanobacterial filaments or colonies were counted. From 30 to 50 filaments or colonies were measured to estimate their biovolume [35] and to determine cyanobacterial biomass by multiplying the number of filaments or colonies by their biovolume. It was assumed that the specific weight of planktonic microalgae is 1.0 g/mL [36] and, therefore, the biomass was expressed in mg/L of water.

The subsample of phytoplankton used for the experiments and the subsamples collected after a seven-day exposure (from all experimental conditions) were preserved with Lugol solution, and then with a formalin and glycerin mixture (3:1). Cyanobacteria and eukaryotic algae were enumerated and their biomass was measured as described above. To determine the *P. agardhii* biovolume for biomass calculation, 50 filaments from each experimental variant were measured. Chlorophyll-a was analysed spectrophotometrically (Specord 40 Analytik Jena). For this purpose, the samples passed through GF/C Whatman filters were extracted for 5 min in a 90% ethanol in water bath at 75 °C according to PN-ISO 10260 [37]. P–PO_4_ and total phosphorus (P tot.) in scum extracts were analysed spectrophotometrically (Specord 40 Analytik Jena) using a method with ammonium molybdate [38]. N-NO_3_ and total nitrogen (N tot.) were analysed using flow analysis method (Fiacompact analyser, MLE) according to DIN ISO 13395 [39] and DIN EN ISO 29441 [40], respectively.

### 2.4. Extraction and HPLC Analysis

Dense bloom samples of *P. agardhii*-dominated biomasses were used to prepare crude extracts for the determination of MCs, and for experiments. The samples were sonicated (for 20 min, 50 W; SONOPULS ultrasonic homogeniser, Bandelin) and after centrifugation (14,000 rpm for 10 min at 17 °C), supernatants were collected. Cyanotoxins were analysed in the extracts before experiments.

Three subsamples of *P. agardhii* from particular experimental variants exposed to the extracts were mixed in order to get one sample with sufficient biomass for toxin analysis. The final volume of the control sample was equal to 70 mL, whereas for the variants with the extracts it was from 15 to 70 mL (depending on *P. agardhii* biomass). Subsamples were filtered onto GF/C glass-fibre filters (Whatman). The extraction of MCs from *P. agardhii* exposed to the extract Pa-A and Pa-B was carried out in 75% (v/v) methanol (Merck, pure p.a.) acidified with 0.01 M HCl, using ultrasonication (two times for 5 min, 50 W; SONOPULS ultrasonic homogeniser, Bandelin). After centrifugation (14,000 rpm for 10 min at 17 °C) and collection of supernatants, filters with the biomass were back-extracted (once for 5 min), and after centrifugation, the combined supernatants were collected in glass tubes and kept at −20 °C until the day of cyanotoxin analysis.

#### HPLC-PDA Analysis of Microcystins

An HPLC-photodiode array detection system (Shimadzu) was used for the detection and preliminary identification of MCs in crude cyanobacterial extracts, and in the extracts of samples used for experiments and collected after a seven-day exposure. The UV detection range was 200–300 nm. [D-Asp^3^]MC-RR, -RR, -YR, -HtyR, [D-Asp^3^]MC-LR, -LW, -LA, -LY, -WR, -LF (Alexis) were used as the standards. Separation of extract components was performed using a Purosphere column (125 × 3 mm, dp 5 μm, Merck) with the mobile phase composed of acetonitrile (Merck): Milli-Q water acidified with 0.05% trifluoroacetic acid (gradient 30–100%), at a flow rate 0.7 mL/min.

### 2.5. Extraction and LC-MS/MS Analysis of Oligopeptides

*P. agardhii* control samples as well as those collected from experiments with the highest concentrations of extracts were mixed in order to get high biomass for oligopeptide analysis. The samples were passed through GF/C glass-fibre filters (Whatman) and freeze-dried. The scum samples used to prepare extracts Pa-A and Pa-B were also lyophilised and subsequently extracted with 2.0 mL of 75% methanol in MilliQ water by a 1 min probe sonication with an ultrasonic disrupter (HD 2070 Sonopuls, Bandeline, Berlin, Germany) followed by a 15 min bath sonication (Sonorex, Bandeline, Berlin, Germany). The samples were then centrifuged at 10,000× *g* for 15 min and the obtained supernatants were subjected to LC-MS/MS analysis, as described in Mazur-Marzec et al. [41]. An Agilent 1200 (Agilent Technologies, Waldboronn, Germany) coupled online to a hybrid triple quadrupole/linear ion trap mass spectrometer (QTRAP5500, Applied Biosystems, Sciex; Concorde, ON, Canada) was used. As a mobile phase, a mixture of A (5% acetonitrile in MilliQ water plus 0.1% formic acid) and B (0.1% formic acid in acetonitrile) was used. Separation was performed on a Zorbax Eclipse XDB-C18 column (4.6 × 150 mm; 5 μm) (Agilent Technologies, Santa Clara, CA, USA). To characterise the structure of cyanobacterial peptides the MS/MS experiments were run using the information dependent acquisition method (IDA) and enhanced ion product mode (EIP). Data acquisition and processing were accomplished using Analyst QS^®^ 1.5.1 software (version 1.5.1, Applied Biosystems, Concord, ON, Canada). The relative content of the particulate peptides in *P. agardhii* after exposure to the extracts were expressed as the ratio of chromatographic peak area to the *P. agardhii* biomass. Based on the analysis of standard MCs and APs, the limit of detection-LOD (at S/N > 3) for the peptides was assessed to be within a range of 0.1–0.5 ng/L.

### 2.6. Data Analysis

The similarities in oligopeptide profiles in the extracts, control and experimental variants with the highest concentration of the extracts were compared using the Jaccard index of similarity [42]. *P. agardhii* biomass and Chl-a concentration were expressed as mean values (*n* = 3) ± standard error (SE). MC concentrations were expressed as mean values (*n* = 3) ± standard deviation (SD). Significant differences among treatments were evaluated using one-way analysis of variance (ANOVA) followed by the Tukey test (*p* < 0.05).

## 3. Results

### 3.1. Characteristics of Two Planktothrix-Dominated Scum Samples and Their Crude Extracts Used in Experiments

Cyanobacterial scum samples used to prepare extracts Pa-A and Pa-B for experiments originated from two different lakes and were predominated by *P. agardhii* (Table 1). Scum samples and extracts were rich in MC isoforms and other oligopeptides (Table 2, Appendix A). The extracts contained similarly high numbers of peptides (50 in Pa-A and 55 in Pa-B), but their profiles were different; only 16% of the identified oligopeptides were the same in both extracts. Generally, aeruginosins (AERs), cyanopeptolins (CPs) and anabaenopeptins (APs) were the most numerous (Appendix A). On the basis of the *m*/*z* values and the fragmentation spectra of the pseudomolecular ions ([M + H]^+^), the structures of the compounds found in both extracts were characterised as microcystin-LR (*m*/*z* 995), dmMC-LR (*m*/*z* 981), [Asp^3^]MC-RR (*m*/*z* 512), two APs (*m*/*z* 858 and *m*/*z* 752), planktopeptin (PLP; *m*/*z* 801), six AERs (*m*/*z* 749, 715, 691, 637, 603 and 575) and two unidentified peptides with *m*/*z* 953 and 685. The extract Pa-A contained higher number of APs (10) and unidentified peptides (11) and lower number of AERs (13) and MCs (4) than the extract Pa-B (5, 9, 19 and 6, respectively). In the extract Pa-B, obtained from the scum sample with a higher contribution (18%) of cyanobacteria other than *P. agardhii* (Table 1), four microginins (MGRs) and one aeruginosamide (AERD) were also detected (Table 2). The extract Pa-A contained approximately 13-fold lower concentrations of MCs than the extract Pa-B (Table 1). In the extract Pa-A, [Asp^3^]MC-RR had the highest contribution, followed by [Asp^3^]MC-HtyrR and dmMC-LR and MC-LR. In the extract Pa-B, MC-RR and MC-RR analogues prevailed, followed by dmMC-LR and MC-LR, then MC-LF.

The extracts were rich in biogenic compounds (Table 1) that were added in different dilutions in the experiments (Figure 1). The extract Pa-B contained slightly higher concentrations of total phosphorus and total nitrogen than the extract Pa-A. Concentrations of P-PO_4_ were two-fold higher, whereas concentrations of N-NO_3_ were ca. three times higher in the extract Pa-B than in the extract Pa-A.

### 3.2. Effect of the Extracts on P. agardhii Biomass and Production of Intracellular MCs

The cyanobacterial extracts obtained from two *P. agardhii*-dominated scum samples with different numbers and concentrations of MC isoforms as well as the diverse composition of other oligopeptides and with a high content of nutrients (Table 1 and Table 2) exerted positive effects on the biomass accumulation (Figure 2), chlorophyll-a (Figure 3) and MC content (Figure 4) in another *P. agardhii*-dominated natural sample. After the seven-day exposure to two extracts, the *P. agardhii* biomass increased several times in comparison with the controls; however, the differences between particular treatments were not statistically significant (Figure 2). *P. agardhii* exposed to the extract Pa-A achieved maximum density 558 mg/L, whereas the density of *P. agardhii* exposed to the extract Pa-B was 637 mg/L. The extract Pa-B contained higher concentrations of both dissolved and total nitrogen and phosphorus (Table 1). The contribution of the *P. agardhii* biomass to the total phytoplankton biomass increased from 96% at the beginning of the experiments (Table 1) to 97.0–99.4% in all experimental conditions (including controls) after the seven-day exposure (Appendix A). The contribution of other cyanobacteria, which was lower than 1%, slightly decreased, except from *Aphanothece* sp. (Appendix A).

The analysis of the dissolved MCs in the water from both experiments with the highest concentrations of the extracts Pa-A and Pa-B (treatment IV) showed that after seven days, the MC content decreased by 50% and 59% in variants with the extracts Pa-A and Pa-B, respectively (in comparison to their initial concentrations). In both experiments, MCs did not inhibit the accumulation of *P. agardhii* biomass, though their total concentration was ca. 13-fold higher in the extract Pa-B than in the extract Pa-A. The content of Chl-a in biomass of the exposed *P. agardhii* (Figure 3) was similar after exposure to both extracts. It ranged from 1.71 to 7.64 µg/mg of *P. agardhii* exposed to the extract Pa-A and from 1.40 to 7.55 µg/mg of *P. agardhii* exposed to the extract Pa-B.

In the presence of both extracts, the content of MCs in *P. agardhii* biomass increased; however, the dose–effect relationship was not strong (Figure 4). The total content of particular MCs in *P. agardhii* was similar after exposure to both extracts; however, slightly wider range of values of MC content was found after exposure to the extract Pa-B (from 0.196 to 0.285 µg/mg of *P. agardhii* biomass) than Pa-A (from 0.203 to 0.241 µg/mg). The content of particular MC isoforms slightly differed after exposure (Table 2, Appendix A). *P. agardhii* exposed to the extract Pa-A contained MC-RR and its two analogues ([Asp^3^, MeSer^7^]MC-RR and [Asp^3^]MC-RR), whereas *P. agardhii* exposed to the extract Pa-B exhibited only [Asp^3^]MC-RR, such as the control *P. agardhii* (Table 2, Appendix A). In both experiments, *P. agardhii* exposed and control contained dmMC-LR. The content of MC-RR analogues was from two- to 10-fold higher than dmMC-LR content and the contribution of both analogues to the total MC content changed with changing extract concentrations (Figure 4). In both experiments, the highest contribution of dmMC-LR was observed in the conditions with extracts dilutions containing 1.3 mg Chl-a/L. dmMC-LR reached 21% and 29% of the total MC content in *P. agardhii* exposed to the extract Pa-A and Pa-B, respectively. Generally, dmMC-LR contribution was slightly higher after exposure to the extract Pa-B (12%–29%) than to Pa-A (9%–21%), containing a much lower concentration of dmMC-LR and MC-LR than the extract Pa-B.

### 3.3. Effect of the Extracts on the Production of Oligopeptides by P. agardhii

The analysis of the similarity in oligopeptide compositions (Table 3) showed that the Jaccard index (J) for oligopeptides in the control *P. agardhii* and in the extract Pa-A, obtained from a cyanobacterial scum sample originating from the same lake as exposed *P. agardhii*, was low and was equal to 0.30. The J index showing the similarity between the control and extract of the biomass from Lake Czarne S. was three-fold lower (J = 0.11). The similarity index for oligopeptides in *P. agardhii* exposed to both extracts, and oligopeptides contained in these extracts, was equal to 0.25 and 0.28 for Pa-A and 0.10 and 0.15 for Pa-B, respectively. The similarity between peptide composition in the control *P. agardhii* and in the exposed *P. agardhii* was equal to 0.38 and 0.49 after exposure to the extracts Pa-A and Pa-B, respectively. The highest similarity in oligopeptide compositions was observed in *P. agardhii* exposed to both extracts (J = 0.60). Moreover, besides dmMC-LR and [Asp^3^]MC-RR, the same oligopeptides, independent of their amounts (Figure 4), were found in the exposed *P. agardhii*: four CPs (*m*/*z* 930 and 916, 879, 865), three APs (*m*/*z* 851, 844, 837), seven AERs (*m*/*z* 759, 749, 715, 691, 637, 635, 583) and one unknown peptide (*m*/*z* 959).

In total, 41 different oligopeptides were detected in the control *P. agardhii* and *P. agardhii* exposed for seven days to the highest concentrations of the extracts (Appendix A, Figure 5A). The total number of peptides in the control was 29, whereas in *P. agardhii* exposed to the extracts, 25 and 23 peptides were detected after exposure to the extracts Pa-A and Pa-B, respectively (Table 2, Appendix A, Figure 5A). The highest difference was observed in the number of unidentified peptides (Appendix A, Figure 5A). Among seven unidentified peptides detected in the control in amounts from 0.10 to 0.64 cps 10^4^/mg of *P. agardhii* biomass (Figure 5A), those with *m*/*z* 1137, 1021, 891 and 885 were not detected in *P. agardhii* exposed to both extracts, although peptides with *m*/*z* 1137 and 1021 were detected in the extract Pa-A (Appendix A). The peptide with *m*/*z* 959, present in the extract Pa-A, had two-fold higher content in *P. agardhii* exposed to this extract than in the control (Figure 5A). Three unidentified peptides (*m*/*z* 959, 953, 929) were found in *P. agardhii* exposed to the extract Pa-B and their content did not change in comparison with the control, although a peptide with *m*/*z* 953 was also detected in the extract Pa-B (Appendix A, Figure 5A). Total number of APs and AERs did not change (Table 2); however, their composition and the relative content in *P. agardhii* biomass differed after exposure (Appendix A, Figure 5A). In *P. agardhii* exposed to the extract Pa-A, the content of APs with *m*/*z* 851, 844, 837, that were also present in the extract Pa-A (Appendix A), increased in comparison with the control. In *P. agardhii* exposed to the extract Pa-B, the content of APs with *m*/*z* 851 and 837, that were not detected in the extract Pa-B (Appendix A), decreased (Figure 5A). In the biomass of *P. agardhii* exposed to the extract Pa-A, one variant of MRG (*m*/*z* 650) of content equal to 0.26 cps 10^4^ mg^−1^ was also found, whereas in *P. agardhii* exposed to the extract Pa-B, high content (0.64 cps 10^4^/mg) of aeruginosamide (*m*/*z* 561) was detected. PLP (*m*/*z* 801) observed in the control in lower content (0.09 cps 10^4^/mg), was not detected in *P. agardhii* exposed to both extracts in which, however, it was detected (Appendix A). In general, among 25 peptides detected in *P. agardhii* exposed to the extract Pa-A, the content of 15 increased in comparison with the control, whereas nine other peptides detected after exposure to the extract were not found in the control (Figure 5A). Among the abovementioned peptides, 12 were present in the extract Pa-A (Appendix A).

The total content of oligopeptides in *P. agardhii* biomass exposed to the extract Pa-A was slightly higher, whereas in *P. agardhii* exposed to the extract Pa-B it was ca. 25% lower than in the control (Figure 5B). Unidentified peptides prevailed in the control in *P. agardhii* exposed to the extract Pa-A, AERs slightly prevailed over CPs, MCs and APs. In *P. agardhii* exposed to the extract Pa-B, a similar contribution of CPs, AERs, APs, MCs and unidentified peptides was found. MRG and AERD that were not found in the control had a low contribution to the total content of oligopeptides in *P. agardhii* exposed to the extracts (Figure 5B).

## 4. Discussion

Harmful cyanobacterial blooms have globally increased in frequency and intensity in recent decades; therefore, cyanobacterial ecology and the bioactive metabolites produced by these organisms warrant more and more attention [10,11,12,17,18]. In this study, we exposed a natural population of *P. agardhii* to two cyanobacterial extracts, obtained from two different cyanobacterial scums formed by this species. Our study showed that in the presence of a high number and diverse composition of bioactive oligopeptides (MCs and other mostly non-ribosomal peptides) and high concentrations of biogenic compounds (nitrogen and phosphorus), the final accumulation of *P. agardhii* biomass and its content of Chl-a and MCs increased. Interestingly, no significant increase in biomass of accompanying cyanobacterial species was observed. Changes in the peptide profiles of the *P. agardhii* community were observed after exposure to two different extracts. As we studied a natural population of the toxigenic *P. agardhii*, we could observe more complex phenomena than in the case of experiments on laboratory strains. In nature, *P. agardhii* assemblages can be composed of several different chemotypes [15].

### 4.1. Higher P. agardhii Productivity: The Role of Nutrient Availability

Although MCs are toxic to many aquatic organisms, including some cyanobacteria and eukaryotic algae [43], our study showed positive effects of the cyanobacterial extracts containing both MCs and other oligopeptides on *P. agardhii* biomass that was not affected by the extracellular MCs even at their broad concentration range (4–410 µg/L). The observed increase in biomass could be a result of the high concentration of nutrients released from the extracted cells that formed cyanobacterial scums. In fact, cyanobacteria contain high amounts of built-in biogenic substances [44]. Nitrogen and phosphorus stimulate cyanobacterial bloom development [14,45]. The results obtained could also be compatible with indirect effects of heterotrophic bacteria present in the natural cyanobacterial assemblages [46]. Therefore, such bacteria could readily decompose the extracts and create environmental conditions more favourable for cyanobacterial development. Our results showed that after a seven-day exposure, the content of MCs in water decreased by 50%–59% in the variants with the highest concentrations of the extracts. The high abundance of oligopeptides coming from the scum extracts could serve as a source of valuable substrates for cyanobacteria. While cyclic MCs may be less prone to bacterial peptidase hydrolysis, other linear peptides, such as aeruginosins (present in cyanobacterial extracts), can likely be utilised [47]. The potential mixotrophic characteristics found in some cyanobacteria [48] could also explain the biomass accumulation and probably the shift in the oligopeptide pattern found in the natural assemblage. Nevertheless, MCs themselves may also exert some positive effects on cyanobacteria colony size. For example, MC-RR exposure caused a significant increase in the production of extracellular polysaccharides (EPS) in *Microcystis*, although it did not influence cyanobacterial growth rate [49]. Moreover, both MC-RR and MC-LR (at concentrations 0.25–10 µg/L) significantly enhanced *Microcystis* colony sizes. Therefore, according to Gan et al. [49], the cellular release of MCs may play a key role in the persistence of cyanobacterial colonies and the dominance of *Microcystis.* There are some other reports that are in accordance with the study by Gan et al. [49], indicating that bioactive cyanobacterial compounds themselves are not essential for cyanobacterial growth [50,51].

Our results showed that the content of Chl-a in the exposed *P. agardhii* in both experiments changed in a similar way independently from broad MC concentration. Moreover, the *Planktothrix* biomass plateaued, while the Chl-a content kept increasing with the increased MC concentration. It may suggest a positive effect of the extracts’ components on photosynthesis efficiency of exposed *P. agardhii*. For example, the concentration of nitrogen in the environment and Chl-a in photosynthetic organisms are directly related [52]. Increased Chl-a production could also be connected with lower light availability caused by cell self-shading and the change of colour of lake water by the extract added. Our results suggest that cyanobacterial metabolites containing biogenic substances may stimulate development of *P. agardhii* in natural cyanobacterial assemblages when water blooms caused by other species collapse [11] with a release of cell content. Future experiments with extracts of algae that cannot produce cyanobacterial oligopeptides, or with similar amounts of available N, P and C-substrate can confirm our findings.

### 4.2. Changes in Content of MCs in the Exposed P. agardhii

The cellular content of MCs in the biomass of *P. agardhii* exposed to the extracts increased in comparison with the controls in both experiments, albeit in different patterns in the case of derivatives of MC-RR and similarly in the case of dmMC-LR. In general, it suggests positive effects of some extracts’ components but not MCs themselves on the MC content in the exposed *P. agardhii*. A recent study [53] showed that the addition of pure MC-LR (10 and 60 μg/L) did not significantly alter expression of mcyB and mcyD responsible for MC production in another cyanobacterium, *Microcystis aeruginosa*. On the other hand, Schatz et al. [29] observed that lysis of *Microcystis* cells, in addition of the MC-LR or micropeptin and microginin, triggered MC production in *Microcystis* through an unknown auto-induction process. Various environmental factors, such as water temperature, light and nutrient concentrations, influence MC synthesis indirectly through cellular growth [19,54]. In general, the toxicity of cyanobacterial blooms depends on the biomass of toxin producers [55,56] and the proportion of toxic genotypes in the population [45]. In our study, it has been confirmed that production of different MC isoforms, mainly derivatives of MC-RR and dmMC-LR, is a characteristic feature of *Planktothrix* subpopulations [17,56,57]. We also found changes in the contribution of particular MC isoforms to the cellular content of MCs produced by *P. agardhii* exposed to different concentrations of the extracts. For example, the contribution of dmMC-LR to the total MC content was the lowest in controls and the highest in experimental variants exposed to the extracts with a Chl-a concentration of ca. 1.3 mg/L. Tonk et al. [58] showed that the total cellular MC content in a *P. agardhii* strain remained constant independent of light intensity; however, the ratio of dmMC-RR to dmMC-LR changed as a response to differing light intensity. Since dmMC-LR is considered to be more toxic than dmMC-RR, the findings of Tonk et al. [58] imply that *P. agardhii* becomes more toxic at high light intensities. On the other hand, our results showed that this species had the highest toxicity potential at optimal growth conditions probably connected with lower light availability caused by cell self-shading and the change of colour of lake water by the extract added. It seems that the ca. two-fold decrease in the contribution of dmMC-LR to the total MC content in the treatments with the highest extract concentrations, in which *P. agardhii* biomass decreased or stopped growing, may suggest that dmMC-LR content might be linked to *P. agardhii* cell density. Agha et al. [55] hypothesised that physiological effects caused by intense light irradiation and/or nutrient limitation, especially N-deficiency, caused an overall reduction in MC cellular concentrations, leading to alterations in the oligopeptide profiles employed for chemotype delimitation.

### 4.3. Changes in Oligopeptide Profiles in P. agardhii Population Exposed to the Extracts

Our experiments may suggest that dissolved oligopeptides present in the extracts did not accumulate in *P. agardhii* under exposure conditions. It seems to be confirmed by the similar numbers of oligopeptides found in the exposed *P. agardhii* and in the control as well as by high similarity in oligopeptide composition between *P. agardhii* exposed to both extracts that differed considerably in oligopeptides composition. The various oligopeptide profiles in three different *P. agardhii*-dominated assemblages found in our study seems to confirm previous findings [15,17] that the populations of bloom-forming *P. agardhii* may consist of several chemotypes of different oligopeptide profiles. We found differences in oligopeptide compositions of *P. agardhii* inhabiting Lake Syczyńskie in two different years. Therefore, we may conclude that the population of *P. agardhii* in that lake was not clonal. Studies dealing with the dynamics of chemotypical subpopulations show differences in terms of chemotype diversity. For instance, Rohrlack et al. [59] identified four *Planktothrix* chemotypes in Lake Steinsfjorden throughout a period of 33 years. Welker et al. [60] found 37 different *Microcystis* chemotypes in Brno reservoir in a study period of five months, although Agha et al. [55] suggested that these values can be overestimated. The relative abundance of dissimilar chemotypes in natural communities, as well as the shifts in chemotypical subpopulations, have been shown to be responsible for the varying MC concentrations observed in Tajo River in Spain [53] and, consequently, have to be considered as critical factors modulating bloom toxicity. The relative abundances of chemotypes in the population are not static and individual subpopulations are subject to strong fluctuations over the season, leading to marked temporal dynamics [55,59]. Our experiments suggest that the changes in oligopeptide composition, indicating changes in the abundance of particular chemotypes in the *P. agardhii* population, may be quick as a response to dissolved compounds (including nutrients) of other cyanobacteria. The changes observed might result from different survivorship of particular chemotypes under experimental conditions. Briand et al. [61] noted that oligopeptides found in the intracellular fractions of *M. aeruginosa* were also detected in the surrounding media; therefore, cyanobacterial cells had direct contact with compounds produced by the same species. However, the changes in cyanobacterial biomass and oligopeptide patterns in the enrichments might also be caused by indirect effects, such as the processing of molecules containing nutrients by the associated microbial assemblages [47].

The ecological role of cyanobacterial oligopeptides remains unknown and is widely discussed in different hypotheses [4,17,24,25,26]. It was observed that strains without MC production contained other structurally related peptides instead [51,56,62]. For example, Sedmak et al. [31] suggested that a possible role of cyanobacterial peptides in the natural environment is the control of cyanobacterial population density. The authors showed that the planktopeptin BL1125, anabaenopeptin B and anabaenopeptin F provoked lysis of different non-axenic *M. aeruginosa* cell lines via the induction of virus-like particles. According to Briand et al. [61], the production of oligopeptide variants by coexisting cells may be regulated throughout intraspecific interactions. We noted the appearance of new peptides ([Asp^3^, MeSer^7^] MC-RR, MC-RR, ARED, some AERs, MRG, some CPLs, AP) and loss of some other, mostly unidentified oligopeptides, in *P. agardhii* exposed to the extracts of other *P. agardii* assemblages containing peptides but also high amounts of biogenic compounds. Kurmayer et al. [12] hypothesised, through toxin and bioactive peptide production, that bloom-forming *Planktothrix* spp. act as niche constructors at the ecosystem scale, possibly resulting in an even higher ability to monopolise resources, positive feedback loops and resilience under stable environmental conditions. Further studies in this field are needed.

## 5. Conclusions

The obtained results showed that, when lysed, *P. agardhii* cells release into the water compounds that may increase the biomass as well as the content of Chl-a and MCs in a natural *P. agardhii* population. MCs present in the extracts did not affect the increase in *P. agardhii* biomass. Although increasing concentrations of *P. agardhii* metabolites did not alter total MC content, changes in the relative contributions of dmMC-LR and MC-RR derivatives were found. No evidence for the bioaccumulation of dissolved oligopeptides in the exposed *P. agardhii* was noted. The similarity between the oligopeptide profiles was highest for *P. agardhii* exposed to each extract. As the two tested extracts differed considerably in oligopeptide composition and contained similar high concentrations of nutrients, it seems that biogenic compounds, not oligopeptides themselves, positively influenced the biomass accumulation of a natural *P. agardhii* population.

## Figures and Tables

**Figure 1 ijerph-17-02881-f001:**
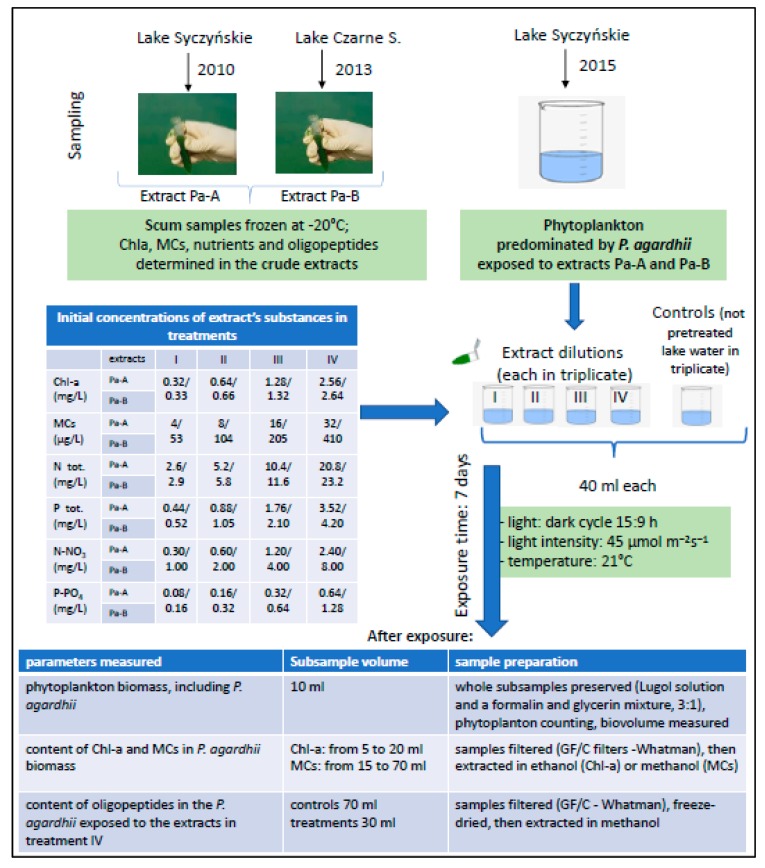
Experimental design and conditions.

**Figure 2 ijerph-17-02881-f002:**
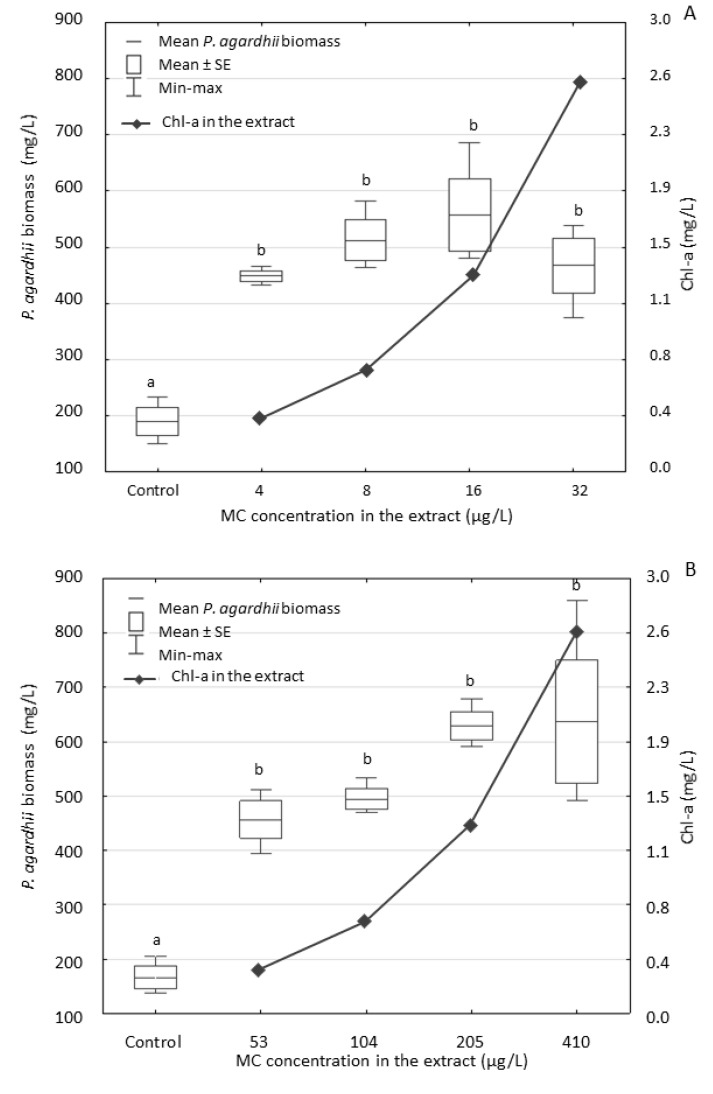
The biomass of *P. agardhii* after the seven-day exposure to the extracts Pa-A (**A**) and Pa-B (**B**) originally containing different concentrations of MCs. Both extracts dilutions had similar concentration of Chl-a. Data are expressed as means ± SE, *n* = 3. Different lowercase letters (a,b) indicate statistically significant differences in *P. agardhii* biomass between experimental treatments (ANOVA, Tukey’s test, *p* < 0.05).

**Figure 3 ijerph-17-02881-f003:**
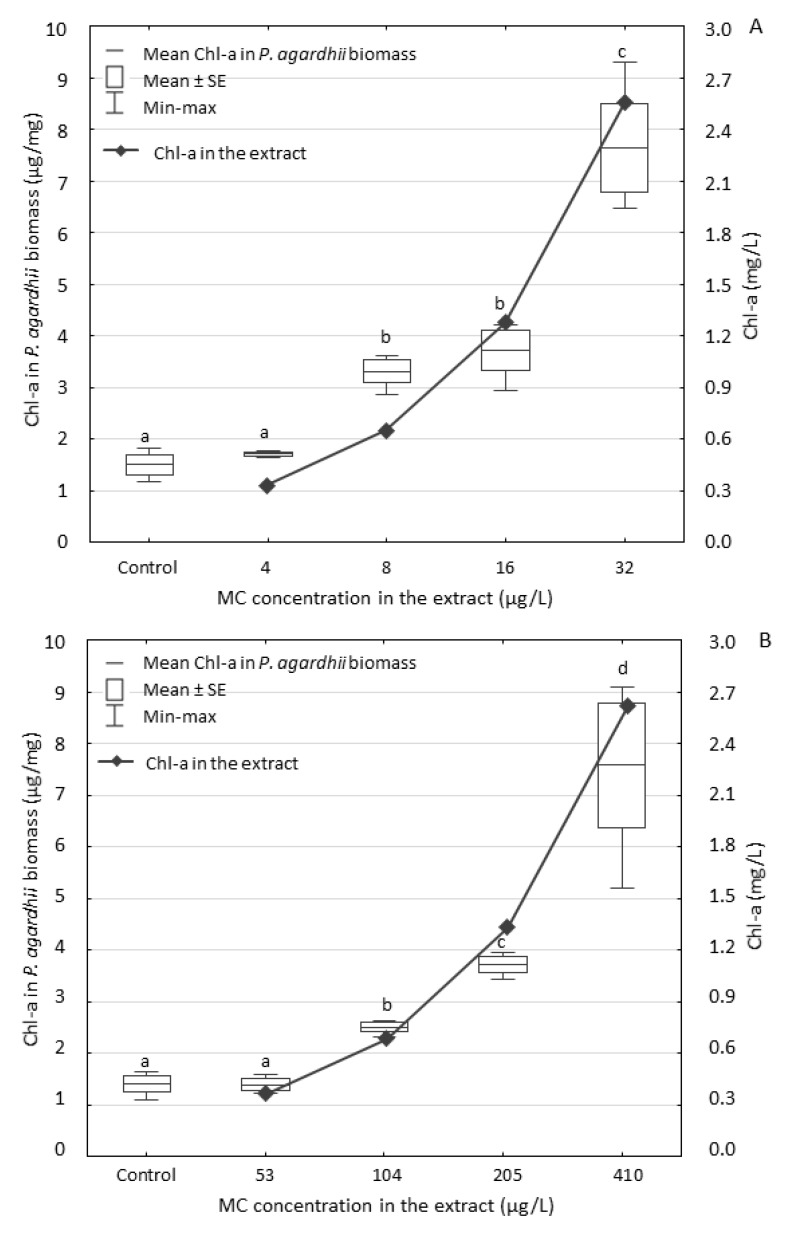
Chl-a content in *P. agardhii* biomass after the seven-day exposure to the extracts Pa-A (**A**) and Pa-B (**B**) originally containing different concentrations of MCs. Both extracts dilutions had similar concentration of Chl-a. Data are expressed as means ± SE, *n* = 3. Different lowercase letters (a–d) indicate statistically significant differences between experimental treatments (ANOVA, Tukey’s test, *p* < 0.05).

**Figure 4 ijerph-17-02881-f004:**
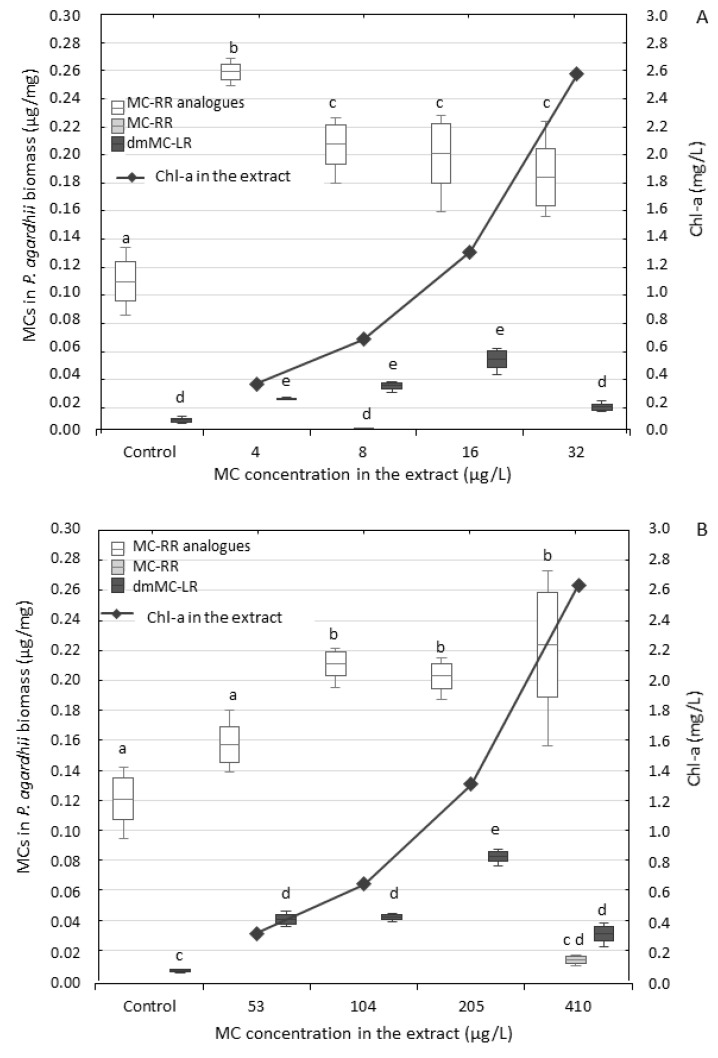
Content of particulate MC isoforms in the biomass of *P. agardhii* after the seven-day exposure to the extracts Pa-A (**A**) and Pa-B (**B**). Data are expressed as means ± 0.5 SD, *n* = 3. Bars means min–max. values. Different lowercase letters (a–d) indicate statistically significant differences between treatments for MC-RR analogues, MC-RR and dmMC-LR (ANOVA, Tukey’s test, *p* < 0.05).

**Figure 5 ijerph-17-02881-f005:**
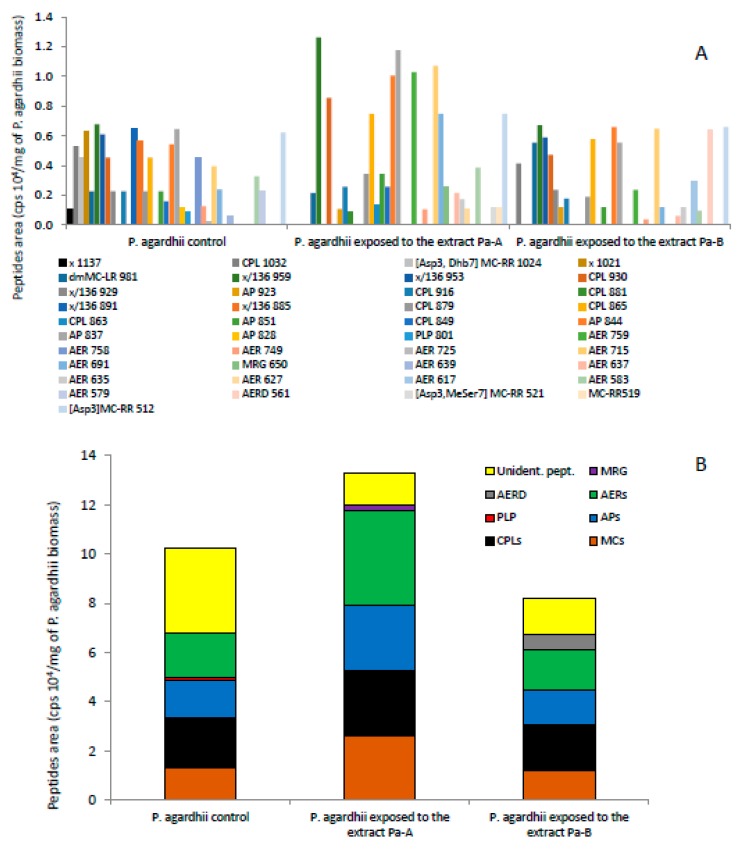
Relative content of particular oligopeptides (**A**) and oligopeptides groups (**B**) in *P. agardhii* exposed to the extracts obtained from two *P. agardhi*-dominated scum samples.

**Table 1 ijerph-17-02881-t001:** Characteristics of two different *Planktothrix*-dominated scum samples, their crude extracts (Pa-A, Pa-B) and the phytoplankton sample with *P. agardhii* used in the experiments. Data are expressed as means ± SD (*n* = 3).

Parameters		Biomasses, Extracts and Samples Used in Experiments (As Controls and Treated)		
Lake, sampling year		Lake Syczyńskie 2010 Pa-A	Lake Czarne S. 2013 Pa-B	Lake Syczyńskie 2015 (conditions correspond to the controls at the beginning of the experiments)
Species contribution to the biomass		***Planktothrix agardhii* (95%)** *Microcystis aeruginosa* (5%)	***P. agardhii* (82%)** *Microcystis spp.* (10%) *Aphanizomenon gracile* (8%)	***P. agardhii* (96%)**, *Planktolyngbya limnetica* (1.4%), other cyanobacteria (0.6%)
Biomass (mg/L)		2800 ± 80	2750 ± 33	105 ± 2
Chl-a (mg/L)		81.86 ± 2.50	80.87 ± 4.1	0.357 ± 0.042
Total MCs (µg/L)		1030 ± 150	13,040 ± 1790	38 ± 2
MCs in *P. agardhii* biomass (µg/mg)		368 ± 54	4748 ± 651	362 ± 19
MCs’ variants (%)	[Asp^3^,Dhb^7^] MC-RR; [Asp^3^]MC-RR; MC-RR	47.3 ^a^	49.0	94.4 ^b^
	[Asp^3^] MC-HtyrR	38.1	n.d.	n.d.
	dmMC-LR; MC-LR	14.6	34.1	5.6 ^c^
	MC-LF	n.d.	16.9	n.d.
P-PO_4_ (mg/L)		20.07 ± 0.52	38.68 ± 0.47	n.det.
P tot. (mg/L)		110.70 ± 10.25	129.37 ± 14.71	0.202
N-NO_3_ (mg/L)		74.63 ± 2.60	250.32 ± 3.37	n.det.
N tot. (mg/L)		641.48 ± 3.57	720.00 ± 3.99	6.843

^a^—only [Asp^3^]MC-RR was detected; ^b^—only two different MC-RR analogues were detected; ^c^—only dmMC-LR was detected; n.d.—not detected; n.det.—not determined.

**Table 2 ijerph-17-02881-t002:** Composition and number of oligopeptides found in two different extracts (Pa-A and Pa-B) of *P. agardhii*-dominated scum samples, and in the biomass of the control *P. agardhii* and *P. agardhii* exposed for seven days to the highest concentrations of the cyanobacterial extracts.

	Extracts		*P. agardhii* Control	*P. agardhii* Exposed to the Extract Pa-A	*P. agardhii* Exposed to the Extract Pa-B
Peptide Classes	Pa-A	Pa-B			
Microcystins (MCs)	4	6	3	4	2
Cyanopeptolins (CPLs)	11	10	6	7	5
Anabaenopeptins (APs)	10	5	4	4	4
Planktopeptin (PLP)	1	1	1	n.d.	n.d.
Aeruginosisns (AERs)	13	19	8	8	8
Aeruginosamide (AERD)	n.d.	1	n.d.	n.d.	1
Microginins (MRGs)	n.d.	4	n.d.	1	n.d.
Unidentified peptides	11	9	7	1	3
Total number	50	55	29	25	23

n.d.—not detected.

**Table 3 ijerph-17-02881-t003:** The Jaccard index of similarity of the composition of oligopeptides found in the cyanobacterial extracts, control *P. agardhii* and *P. agardhii* exposed to the extracts.

Compared Samples	Extract Pa-B	*P. agardhii* Control	*P. agardhii* Exposed to the Extract Pa-A	*P. agardhii* Exposed to the Extract Pa-B
Extract Pa-A	0.16	0.30	0.25	0.28
Extract Pa-B	-	0.11	0.10	0.15
*P. agardhii* Control	-	-	0.38	0.49
*P. agardhii* exposed to the extract Pa-A	-	-	-	0.60

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
