# Peer review of "The Effects of Cyanobacterial Bloom Extracts on the Biomass, Chl-a, MC and Other Oligopeptides Contents in a Natural *Planktothrix agardhii* Population"

_ijerph, 2020, doi:10.3390/ijerph17082881_

Round 1
Reviewer 1 Report
General comments
This work tested the effects of cyanobacterial bloom extracts on the biomass, Chl-a and MC accumulation of a natural population dominated by Planktothrix agardhii. This issue is not reflected in the title.
My main concerns are which question the authors want to address and the experimental design that they chose. The question is not clear, the experiments are based in extremely complex material, a natural population of cyanobacteria dominated by P. aghardii incubated with bloom extracts. Thus, it is difficult to isolate the effect of nutrients from the effect of bioactive oligopeptides or else, other molecules from the diverse microbial community present. It is also difficult to characterize physiological responses to the extracts because the experiments are performed with a complex mixture of strains. The control is poorly described and compromises the interpretation of the results, as detailed below. The results are presented in a confusing way and are not correctly interpreted. The conclusions and discussion are not supported by the results.
Title
The title refers to "toxigenic cyanobacterium Planktothrix agardhii" which indicates a focus in microcystins (MCs) but in contrast, the text explores the importance of several oligopeptides.
After examining the experiments I do not consider that the study addresses "Intraspecific relationships" since the authors test the effects of cell extracts of natural populations on another natural population of P agardhii. See last paragraph of introduction.
Also, “natural chemotypes” were not characterized, instead extracts with different compositions of oligopeptides were used.
Abstract
The abstract is confusing and the authors did not present the aim or hypothesis of the work.
ln 16 "We analysed the growth, chlorophyll-a production and oligopeptide composition of a natural population of P. agardhii after a 7-day exposure to crude extracts of two different P. agardhii-dominated bloom samples."
growth -> biomass accumulation
chlorophyll-a production -> Chl content
ln 27 "Our results suggest that compounds coming from the decomposed oligopeptides and other metabolites may play a role in the regulation of the growth of different P. agardhii chemotypes"
The study did not investigate the stability or decomposition of oligopeptides.
The study did not analyze "the growth of different P. agardhii chemotypes" individualy, it analyzed the biomass accumulation of a mixed natural population, and no data related to "regulation of growth" was presented.
Ln 55 “Peptides comprise more than 60% of the known biologically active compounds produced by cyanobacteria ...current research is strongly focused on exploring new oligopeptide variants and their bioactive properties, the biological role of these compounds remains elusive “
Exploring the bioactive properties of oligopeptides is an interesting issue, but when you test oligopeptides in the form of a complex extract it is very difficult to attribute any effect to these molecules. In my view, in this condition, oligopeptides become part of an immense pool of generic peptides originating from degraded proteins. Thus, the nutritional effect becomes predominant. That is, the effect of the oligopeptides or their bioactivity would have to be demonstrated with a fraction enriched in these compounds, or with purified oligopeptides.
ln 77 The authors describe the aim of this work but it is not related to the title. Similarly, the discussion adds a lot of information not related to this objective.
Ln 80 “We hypothesised that the growth and production of oligopeptides by P. agardhii may be shaped by biogenic compounds and/or secondary metabolites, including toxic MCs, contained in other P. agardhii-dominated cyanobacterial assemblages.”
There is a problem in the experimental design, it is not possible to distinguish the effect of biogenic compounds from that of secondary metabolites.
Methods
Important information is missing:
ln 85-91 These extracts were collected 5 and 3 years before being used in the experiments.
How they were stored?
ln 99 “experiments were performed for seven days in the lake water”
Which lake, which date of collection? The authors should explain how this lake water was treated, filtered, stocked, etc. How about the nutrient composition of this lake water? The control condition was not clearly described. Table 2 included the control but Table 1 did not.
ln 105 initial content of Chl-a in P. agardhii biomass was 3.4 mg−1 fresh weight (FW)
Chl-a content expressed as 3.4 mg−1 fresh weight (FW)?
How fresh weight was determined? In table 1 Cha is presented as mg L-1.
ln 107 final concentration of biologically active MCs present in the extracts Pa-A and Pa-B
Why “biologically active MCs”? Did you test their biological activity?
Ln 106-108 the content of MCs was 0.362 μg mg−1 FW in P. agardhii biomass and MCs in the extracts 4 to 32 μg l−1
It is difficult to compare MC concentrations in different units. Also, μg mg−1 FW is vague because we do not know how FW was determined. In Table 1 different units are presented.
When did the authors determined the concentrations of Chla, MC and oligopeptides in the extracts? After collection or before use? These extracts were collected 5 and 3 years before being used in the experiments.
Ln 118 Cell biovolume was measured to determine cyanobacterial biomass expressed as fresh weight (FW).
It is confusing to follow the results with measures in FW. Data should be related to biovolume of cyanobacteria (P aghardii).
Table 1. Characteristics of two different Planktothrix-dominated scum samples, their crude extracts (Pa-215 A, Pa-B)
In Methods ln 99: Four dilutions of the extracts of the similar final concentration range of Chl-a (as an equivalent of cyanobacterial biomass), i.e. 0.33–2.58 mg 100 l−1 in the extract Pa-A and 0.33–2.63 mg l−1 in Pa-B, were used in the experiments.
These experimental conditions (the 4 dilutions) as well as the control conditions were not presented. The authors should present them in another table with the resulting values of Chl-a, MC, nutrients etc that resulted in each condition (dilution and extract) tested and the control condition. A complication is that the authors mix a phytoplankton sample with P. agardhii with extracts, and we do not know in which proportion it was done, So, after the mixture of this sample with the extracts which are the resulting values of the cited parameters (biomass, Ch-a, MC, nutrients etc). A basic question: does the addition of the extracts contribute significantly to the resulting biomass/ FW? For exmaple, in the higher concentration described for the extracts added (2.5 mg L-1 Chl-a) these extracts would contribute with amounts of biomass, MC and Chl-a similar to those described for the P. aghardii live sample. This interference will confound the results.
In sum, the experiments are not appropriately described.
Results
The introductory paragraph is unnecessary.
3.1
The data were included in Tables and should not be described in the text.
Table 2 could present MC and oligopeptides composition and abundances.
Ln 220 “The extracts were rich in biogenic compounds (Table 1). The extract Pa-B contained slightly higher concentrations of total phosphorus and total nitrogen than the extract Pa-A.”
How about the control condition? The study should include one control for each nutrient (at least N and P) condition represented by these extracts. Otherwise, we cannot rule out that the observed effects are due to nutrients.
3.2. Effect of the extracts on P. agardhii growth and production of intracellular MCs
ln 229 “extracts obtained from two P. agardhii dominated scum samples with different numbers and concentrations of MC isoforms as well as the diverse composition of other oligopeptides (Tables 1, 2) exerted positive effects on the growth (Fig. 1), chlorophyll-a (Figure 2) and MCs production (Figure. 3) of another P. agardhii. “
Although the extracts contained a diverse composition of other oligopeptides, they could have led to biomass accumulation because they were a rich source of nutrients. Unless the control had the same nutritional characteristics, which was not demonstrated. Looking at fig 1, even at the lower concentration (< 0.5 mg/L Chl-a) the extracts led to biomass increase in comparison to control and the effect was maximum and did not increase in higher additions of extract.
Note that using 0.3 mg/L of Chl-a the extracts were diluted around 250 x and still contained high concentrations of N and P. Also, we do not know if there were other sources of nutrients from the lake water, which was not characterized.
From fig 1 it is apparent that the concentration of MC in the extracts had no effect on biomass accumulation.
Also, no difference in biomass accumulation was observed for the two extracts with diverse composition of other oligopeptides.
“of another P. agardhii. “
This makes no sense because the extracts were obtained from a diverse P. agardhii population and were tested on a diverse P. agardhii population.
Fig 1 biomass after 7 day exposure
It would be more informative to present these parameters at time 0 and after 7 days, increase in biomass. Legend “bioactive MC” ( ?)
ln 244 “MCs, which concentration was ca. 13-fold higher in the extract Pa-B than in the extract Pa-A, did not inhibit the P. agardhii growth”
After 7 days (probably well before) of incubation with a cellular extract and no asseptic conditions, MC originally present in the extracts probably has been degraded by bacteria - as mentioned by the authors (ln 362).
Item 3.2 the authors compare the results of treatments with extracts A and B but not with control.
Figs 2 and 3
When Chl-a or MC contents were determined, which was the contribution of the amount of these molecules in the extracts used to the final values?
Figs 1, 2 and 3, the addition of of extracts resulted in higher P. aghardii biomass, thus, higher chl-a, thus higher MC. The normalization of Chl-a and MC by FW is problematic since it does not show higher cellular quotas.
Also, no clear dose – response effect can be observed for the extract additions.
Thus, the authors can not state that ““extracts obtained from two P. agardhii dominated scum samples exerted positive effects on the growth (Fig. 1), chlorophyll-a (Figure 2) and MCs production”, because the effect was in biomass increase and the other effects were consequences of this higher biomass.
Table 3 The authors did not specify to which extract dilution these results correspond.
Ln 264 the authors describe results from fig 3, “In the presence of both extracts, production of MCs by P. agardhii and the production of particular MC isoforms”.
Since these results were obtained after 7 days of incubation, these differences could be attributed to diverse compositions of P. aghardii populations in each treatment. Upon incubation in laboratory conditions, a natural population is expected to change in the relative abundance of its strains. It will be submitted to selection and also to random fluctuations. This will reflect in the oligopeptide composition of the culture. We do not know if this is the effect of time (7 days) or the effect of the extracts, since we do not have the initial time to compare. This observation also applies to results presented in item 3.3, figs 4 and 5.
As stated by the authors in ln 44: “ development of particular toxic and/or non-toxic subpopulations within cyanobacteria is impossible to predict. Also, among P. agardhii populations, both MC-producing and non-MC-producing strains occur, and the genotypic variation concerns both MCs and other biologically active oligopeptides...the populations of P. agardhii are not clonal and may be composed of several chemotypes.”
No dose- effect relationship was apparent in fig 3.
Also, we do not know about “production “, abundance is a better term.
Table 4 similarity of the composition of oligopeptides found in the cyanobacterial extracts, control P. agardhii and P. agardhii exposed to the extracts
A hetamap would be more informative, also with data from fig 4. The replicates should be presented individually.
Ln 299-323 If data are presented in a figure or table it is unnecessary to describe in the text.
Discussion
ln 340-352 This paragraph concentrates much of my concerns about this study.
“ in the presence of cyanobacterial extracts, obtained from two different cyanobacterial scums formed by P. agardhii and containing a high number and diverse composition of bioactive oligopeptides (MCs and other mostly non-ribosomal peptides) and high concentrations of biogenic compounds (nitrogen and phosphorus) the growth of another P. agardhii population and its content of Chl-a and MCs increased. “
There is no convincing evidence showing that the increase in P. agardhii biomass was not only the consequence of the biogenic compounds (nitrogen and phosphorus). As mentioned by the authors (ln 359). Control condition poorly described.
There is no convincing evidence that content of Chl-a and MCs increased per cell.
There is no data on growth, only the final accumulation of P. agardhii biomass.
“Changes in the peptide profiles of the P. agardhii community were observed after exposure to two different extracts. “
As mentioned before, it could not be distinguished from temporal fluctuation in the population incubated in laboratory conditions. No data corresponding to the initial time is presented.
“ As we exposed a natural population of the toxigenic P. agardhii to the extracts of natural cyanobacterial biomasses predominated (82 and 95%) by this species and containing a little admixture (5 and 18%) of other cyanobacteria, we could observe more complex phenomena, resulting both from intra- and interspecific interactions, than in the case of experiments on individual laboratory strains. “
The experimental design combines two complex materials, a natural population of P. agardhii and extracts of natural cyanobacterial communities containing P. agardhii and other cyanobacteria. Thus, it is extremely difficult to establish a causal relationship between extract composition and physiological response. Although the title refers to intraspecific interactions, this issue is not covered here using extracts.
ln 363 “The high abundance of oligopeptides coming from the scum extracts can have served as valuable substrates for cyanobacterial growth, especially, when nutrients become more limiting after a few days of exposure. “
Nutrient limitation was not shown in the results. The proportion of oligopeptides to total protein or N containing molecules is so small that they would not supply nutrient deficiency.
Ln 374 “Our experiments showed that the content of Chl-a in the exposed P. agardhii correlated with increasing extract concentrations independently from broad MC concentration.”
No correlation was clearly shown in the results.
Ln 376 ”Moreover, the Planktothrix biomass plateaued while the Chl-a content kept increasing with the increased MC concentration.”
Also, not clear from results.And, as stated by the authors below, this can be the effect of turbidity, that was not monitored.
Ln 379 “Moreover, the Planktothrix biomass plateaued while the Chl-a content kept increasing with the increased MC concentration.”
Here is another possible limitation argued by the authors, the resulting culture conditions after the addition of the extracts to P. aghardii cultures was not detailed, as mentioned before. It is expected that the extracts resulted in higher turbidity in comparison to control, thus another variable was introduced in the experiment. This is mentioned again in ln 413.
ln 381 “Our results strongly suggest that dissolved P. agardhii metabolites and other cell compounds containing biogenic substances allow efficient growing and, therefore, domination of P. agardhii in natural cyanobacterial assemblages when water blooms caused by other species collapse.”
I strongly disagree. The effect can be due to nutrient input, this would happen in the collapse of any cyanobacterial bloom.
ln 385 “Future experiments with ... similar amounts of available N, P and C-substrate can confirm our findings.”
In fact, this should have been your control in this study.
item 4.2 there is a confusion in the text to explain a possible effect of extracts on MC amounts. The term “production” is misleading because there was no experiment to measure this, only MC amount. Also, MC quotas were not determined. As the study deals with a natural population, it is a mixture of MC+ and MC- strains, and their contribution is unknown, and can change with the incubation period. So, it is extremely difficult to discuss the effect of the extracts on MC production. Finally, MC content will increase simply as a result of biomass increase.
Ln 423 “dissolved oligopeptides present in the extracts did not accumulate in P. agardhii under exposure conditions “
It is surprising to find this discussion since this was not presented as an objective of this study and experiments were not designed to test this.
Ln 441 “Our experiment showed that the changes in oligopeptide composition, suggesting changes in abundance of particular chemotypes in the P. agardhii population, may be quick as a response to dissolved compounds of other chemotypes of this species or other cyanobacterial species”
There is no evidence in the results to support the statement that changes in abundance of particular chemotypes in the P. agardhii population (that were not characterized) may be a response to dissolved compounds of other chemotypes (that were not characterized either), and in this study these compounds were a minor component of a rich nutritional and complex extract.
Ln 447 “our findings suggest that in natural conditions cyanobacterial metabolites that are released to the water during bloom formation or collapse, may play some role in the regulation of cyanobacterial development at the chemotype level.”
Same observation as above.
Although all the discussion is highly speculative and does not relate to the correct interpretation of the results presented, item 4.4 is specially so.
Author Response
Response to the comments of Reviewer 1
Thank you very much for all valuable comments and suggestions. All were taken into consideration.
General comments
- This work tested the effects of cyanobacterial bloom extracts on the biomass, Chl-a and MC accumulation of a natural population dominated by Planktothrix agardhii. This issue is not reflected in the title.
Response: lines 2-4. The title was changed to The effects of cyanobacterial bloom extracts on the biomass, Chl-a, MC and other oligopeptides contents in a natural Planktothrix agardhii population”.
- My main concerns are which question the authors want to address and the experimental design that they chose. The question is not clear, the experiments are based in extremely complex material, a natural population of cyanobacteria dominated by P. aghardii incubated with bloom extracts. Thus, it is difficult to isolate the effect of nutrients from the effect of bioactive oligopeptides or else, other molecules from the diverse microbial community present. It is also difficult to characterize physiological responses to the extracts because the experiments are performed with a complex mixture of strains. The control is poorly described and compromises the interpretation of the results, as detailed below. The results are presented in a confusing way and are not correctly interpreted. The conclusions and discussion are not supported by the results.
Response: We clarified the unclear issues and methods, results and discussion have been re-written and improved. Thank you
- Title
The title refers to "toxigenic cyanobacterium Planktothrix agardhii" which indicates a focus in microcystins (MCs) but in contrast, the text explores the importance of several oligopeptides.
After examining the experiments I do not consider that the study addresses "Intraspecific relationships" since the authors test the effects of cell extracts of natural populations on another natural population of P agardhii. See last paragraph of introduction.
Also, “natural chemotypes” were not characterized, instead extracts with different compositions of oligopeptides were used.
Response: lines 2-4. The title was changed to reflect a real content of the work. “The effects of cyanobacterial bloom extracts on the biomass, Chl-a, MC and other oligopeptides contents in a natural Planktothrix agardhii population”. Thank you
- Abstract
The abstract is confusing and the authors did not present the aim or hypothesis of the work.
Response: Lines 19-25. The abstract was improved. The aim of the work and the hypothesis were added. There is: “The aim of this work was to study the effects of the extracts of P. agardhii-predominated bloom samples characterised by different oligopeptide profiles and high concentration of dissolved and total nitrogen and phosphorus on another natural P. agardhii population. For this purpose, the biomass, chlorophyll a and oligopeptides content in the treated P. agardhii were measured. We hypothesised that the biomass and production of oligopeptides by P. agardhii may be shaped in a different manner by diverse mixtures of metabolites of different P. agardhii-dominated cyanobacterial assemblages.”
- ln 16 "We analysed the growth, chlorophyll-a production and oligopeptide composition of a natural population of P. agardhii after a 7-day exposure to crude extracts of two different P. agardhii-dominated bloom samples."
growth -> biomass accumulation
chlorophyll-a production -> Chl content
Response: Lines 22-23. The abstract was rewritten and the present sentence is “For this purpose, the biomass, chlorophyll a and oligopeptides content in the treated P. agardhii were measured”. We changed the terms according the reviewer suggestion. Thank you
- ln 27 "Our results suggest that compounds coming from the decomposed oligopeptides and other metabolites may play a role in the regulation of the growth of different P. agardhii chemotypes"
The study did not investigate the stability or decomposition of oligopeptides.
The sentence was rewritten: “It seems that biogenic compounds, not oligopeptides themselves, may play a role in the development of particular P. agardhii chemotypes”.
The study did not analyze "the growth of different P. agardhii chemotypes" individualy, it analyzed the biomass accumulation of a mixed natural population, and no data related to "regulation of growth" was presented.
Response: lines 38-39. We agree. Thanks for advice. The description was changed according to the reviewer suggestion. “It seems that biogenic compounds, not oligopeptides themselves, positively influenced on a mixed natural P. agardhii population”.
- Ln 55 “Peptides comprise more than 60% of the known biologically active compounds produced by cyanobacteria ...current research is strongly focused on exploring new oligopeptide variants and their bioactive properties, the biological role of these compounds remains elusive “
Exploring the bioactive properties of oligopeptides is an interesting issue, but when you test oligopeptides in the form of a complex extract it is very difficult to attribute any effect to these molecules. In my view, in this condition, oligopeptides become part of an immense pool of generic peptides originating from degraded proteins. Thus, the nutritional effect becomes predominant. That is, the effect of the oligopeptides or their bioactivity would have to be demonstrated with a fraction enriched in these compounds, or with purified oligopeptides.
Response: line 70. We agree with this opinion. Our work is the preliminary study and gives the opportunity to continue and broaden studies in this topic.
- ln 77 The authors describe the aim of this work but it is not related to the title. Similarly, the discussion adds a lot of information not related to this objective.
Response: line 93-95. The title was changed, the discussion was improved. The description of the aim of work was Improved: “The aim of this work was to study the effects of the extracts of P. agardhii-predominated bloom samples characterised by different oligopeptide profiles and high concentration of dissolved and total nitrogen and phosphorus on another natural P. agardhii population. “
- Ln 80 “We hypothesised that the growth and production of oligopeptides by P. agardhii may be shaped by biogenic compounds and/or secondary metabolites, including toxic MCs, contained in other P. agardhii-dominated cyanobacterial assemblages.”
There is a problem in the experimental design, it is not possible to distinguish the effect of biogenic compounds from that of secondary metabolites.
Response: Line 95-97. The hypothesis was changed to “We hypothesised that the P. agardhii biomass and content of oligopeptides in P. agardhii may be shaped in a different manner by diverse mixtures of metabolites of different P. agardhii-dominated cyanobacterial assemblages.”
Methods
- Important information is missing:
ln 85-91 These extracts were collected 5 and 3 years before being used in the experiments.
How they were stored?
Response: lines 114-115: The information was added here. “After collection and phytoplankton enumeration (described in point 2.3), the scum samples were frozen (at -20°C) until the day of extraction, analyses and experiments”. Previously it was in line 154:
- ln 99 “experiments were performed for seven days in the lake water”
Which lake, which date of collection? The authors should explain how this lake water was treated, filtered, stocked, etc. How about the nutrient composition of this lake water? The control condition was not clearly described. Table 2 included the control but Table 1 did not.
Response: lines 124-131. The description was improved: “The crude lake water with the natural phytoplankton community from Lake Syczyńskie, predominated by P. agardhii (96% of the total phytoplankton biomass) was exposed after collection (the same day) to the crude cyanobacterial extracts Pa-A and Pa-B obtained from P. agardhii-dominated scum samples collected in different years and lakes (Table 1). At the beginning of the experiments, the composition and abundance of P. aghardii population in each experimental condition was uniform as we used one and the same water sample that was mixed up for homogeneity before setting up experiments. The experiments were performed for seven days.”
Description of Table 1 was improved: “conditions corresponds to the controls at the begining of the experiments” was added in the last column. Both Table 1 and Table 2 contain the information about the controls.
- ln 105 initial content of Chl-a in P. agardhii biomass was 3.4 mg−1 fresh weight (FW)
Chl-a content expressed as 3.4 mg−1 fresh weight (FW)?
How fresh weight was determined? In table 1 Cha is presented as mg L-1.
Response: line 142. The mistake was corrected. Now there is 3.4 µg mg. We express the biomass of P. agardhii in mg. The confusing term “Fresh weight” was deleted. Whereas the description how the phytoplankton biomass, including P. agardhii biomass was calculated was improved (lines 155-157 “Cell biovolume was measured to determine cyanobacterial biomass [35]. It was assumed that the specific weight of planktonic microalgae is 1.0 g ml-1 [36] and, therefore, the biomass was expressed in mg”) .
- ln 107 final concentration of biologically active MCs present in the extracts Pa-A and Pa-B
Why “biologically active MCs”? Did you test their biological activity?
Response: line 134. This confusing term was removed.
- Ln 106-108 the content of MCs was 0.362 μg mg−1 FW in P. agardhii biomass and MCs in the extracts 4 to 32 μg l−1
It is difficult to compare MC concentrations in different units. Also, μg mg−1 FW is vague because we do not know how FW was determined. In Table 1 different units are presented.
Response: We present these values in different states of matter: firstly: MCs per volume of the water in experiments (μg l−1) and secondly MCs per P. agardhii biomass (μg mg-1). Therefore, we use different units. We ask for permission to leave the biomass in mg not in volume as during calculation of the biomass it was assumed that the specific weight of planktonic microalgae is 1.0 g ml-1 [Golterman et al. 1968] so the results presented in mg and mm3 are 1:1. The confusing term “Fresh weight” was deleted. Description how the phytoplankton biomass, including P. agardhii biomass was calculated was improved (lines 161-163). Units in Table 1 were unified and are the same like in the text and Figures.
- When did the authors determined the concentrations of Chla, MC and oligopeptides in the extracts? After collection or before use? These extracts were collected 5 and 3 years before being used in the experiments.
Response: lines 136-137. The information was added: “The concentrations of Chla, MCs and oligopeptides in the extracts were determined before use in the experiments.”
- Ln 118 Cell biovolume was measured to determine cyanobacterial biomass expressed as fresh weight (FW).
It is confusing to follow the results with measures in FW. Data should be related to biovolume of cyanobacteria (P aghardii).
Response: We improved the description of biomass determination (lines 161-163). Confusing term FW was deleted. We suggest to leave the units and result in a present form.
- Table 1. Characteristics of two different Planktothrix-dominated scum samples, their crude extracts (Pa-215 A, Pa-B)
In Methods ln 99: Four dilutions of the extracts of the similar final concentration range of Chl-a (as an equivalent of cyanobacterial biomass), i.e. 0.33–2.58 mg 100 l−1 in the extract Pa-A and 0.33–2.63 mg l−1 in Pa-B, were used in the experiments.
These experimental conditions (the 4 dilutions) as well as the control conditions were not presented. The authors should present them in another table with the resulting values of Chl-a, MC, nutrients etc that resulted in each condition (dilution and extract) tested and the control condition. A complication is that the authors mix a phytoplankton sample with P. agardhii with extracts, and we do not know in which proportion it was done, So, after the mixture of this sample with the extracts which are the resulting values of the cited parameters (biomass, Ch-a, MC, nutrients etc). A basic question: does the addition of the extracts contribute significantly to the resulting biomass/ FW? For exmaple, in the higher concentration described for the extracts added (2.5 mg L-1 Chl-a) these extracts would contribute with amounts of biomass, MC and Chl-a similar to those described for the P. aghardii live sample. This interference will confound the results.
In sum, the experiments are not appropriately described.
Response: lines 124-146. The experiments were appropriately described. Additional scheme showing the experimental design (suggested also by the Reviewer 3) was added (Figure 1). Control conditions are better described and are presented in Table 1.
Results
- The introductory paragraph is unnecessary.
Response: The introductory paragraph was deleted.
- 1. The data were included in Tables and should not be described in the text.
Response: Part of the data described in Tables was deleted from the paragraph.
- Table 2 could present MC and oligopeptides composition and abundances.
Response: We changed the title to “Composition and number of oligopeptides found in two different extracts (Pa-A and Pa-B) of P. agardhii-dominated scum samples, and in the biomass of the control P. agardhii and P. agardhii exposed for seven days to the highest concentrations of the cyanobacterial extracts.”
- Ln 220 “The extracts were rich in biogenic compounds (Table 1). The extract Pa-B contained slightly higher concentrations of total phosphorus and total nitrogen than the extract Pa-A.”
How about the control condition? The study should include one control for each nutrient (at least N and P) condition represented by these extracts. Otherwise, we cannot rule out that the observed effects are due to nutrients.
Response: We did not add any nutrients in control variants of experiments. The controls used were just lake water from Syczynskie Lake sampled in 2015, left without the addition of any substances. The aim of this work was to study the effects of mixtures of cyanobacterial metabolites on a natural population of P. agardhii.
- 2. Effect of the extracts on P. agardhii growth and production of intracellular MCs
ln 229 “extracts obtained from two P. agardhii dominated scum samples with different numbers and concentrations of MC isoforms as well as the diverse composition of other oligopeptides (Tables 1, 2) exerted positive effects on the growth (Fig. 1), chlorophyll-a (Figure 2) and MCs production (Figure. 3) of another P. agardhii. “
Although the extracts contained a diverse composition of other oligopeptides, they could have led to biomass accumulation because they were a rich source of nutrients. Unless the control had the same nutritional characteristics, which was not demonstrated. Looking at fig 1, even at the lower concentration (< 0.5 mg/L Chl-a) the extracts led to biomass increase in comparison to control and the effect was maximum and did not increase in higher additions of extract.
Note that using 0.3 mg/L of Chl-a the extracts were diluted around 250 x and still contained high concentrations of N and P. Also, we do not know if there were other sources of nutrients from the lake water, which was not characterized.
Response: We agree. However, the source of nutrients contained in the from lake water are ischaracterised in Table 1.
- From fig 1 it is apparent that the concentration of MC in the extracts had no effect on biomass accumulation.
Also, no difference in biomass accumulation was observed for the two extracts with diverse composition of other oligopeptides.
Response: Yes, we highlight that the concentration of MC in the extracts had no effect on biomass accumulation (Fig 2).
- “of another P. agardhii. “
This makes no sense because the extracts were obtained from a diverse P. agardhii population and were tested on a diverse P. agardhii population.
Response: But the content of the tested extracts were quite different and could exert different effects, which is our hypothesis. We think that the presumed role of the detected metabolites cannot be ruled out, as the bioactivity of the compounds is well documented.
- Fig 1 biomass after 7 day exposure
It would be more informative to present these parameters at time 0 and after 7 days, increase in biomass. Legend “bioactive MC” ( ?)
Response: The conditions at time 0 are presented in Table 1. “Bioactive” was removed.
- ln 244 “MCs, which concentration was ca. 13-fold higher in the extract Pa-B than in the extract Pa-A, did not inhibit the P. agardhii growth”
After 7 days (probably well before) of incubation with a cellular extract and no asseptic conditions, MC originally present in the extracts probably has been degraded by bacteria - as mentioned by the authors (ln 362).
Response: We decided to add the information about the results of a control analysis of dissolved MCs in the water from variants with the highest concentrations of the extracts. The analysis was carried out after the 7-day exposure only in these two variants but it showed that after 7 days MCs were degraded in 50-59% (lines 151-153 and 293-296). Although MCs may be degraded by bacteria, the ability to break down MCs or structurally similar NOD is not a common feature of bacteria (Dziga et al. 2017, Toruńska-Sitarz 2018).
- Item 3.2 the authors compare the results of treatments with extracts A and B but not with control.
Response: line 274-301. We compared the results with control.
- Figs 2 and 3
When Chl-a or MC contents were determined, which was the contribution of the amount of these molecules in the extracts used to the final values?
Response: Both parameters are shown on Figures.
- Figs 1, 2 and 3, the addition of of extracts resulted in higher P. aghardii biomass, thus, higher chl-a, thus higher MC. The normalization of Chl-a and MC by FW is problematic since it does not show higher cellular quotas.
Also, no clear dose – response effect can be observed for the extract additions.
Thus, the authors can not state that ““extracts obtained from two P. agardhii dominated scum samples exerted positive effects on the growth (Fig. 1), chlorophyll-a (Figure 2) and MCs production”, because the effect was in biomass increase and the other effects were consequences of this higher biomass.
Response: Please note, that the results are shown per mg of P. agardhii biomass not per liter of water and dose -response effect is clearly visible, especially, that the differences among most results are statistically significant. Generally, there are factors that will affect the increased / decreased production of cyanobacterial compounds by specific biomass what was discussed (paragraph 4.1.)
- Table 3 The authors did not specify to which extract dilution these results correspond.
Response: These results correspond to controls and all treatments what is described in row 2 of the table.
- Ln 264 the authors describe results from fig 3, “In the presence of both extracts, production of MCs by P. agardhii and the production of particular MC isoforms”.
Since these results were obtained after 7 days of incubation, these differences could be attributed to diverse compositions of P. aghardii populations in each treatment. Upon incubation in laboratory conditions, a natural population is expected to change in the relative abundance of its strains. It will be submitted to selection and also to random fluctuations. This will reflect in the oligopeptide composition of the culture. We do not know if this is the effect of time (7 days) or the effect of the extracts, since we do not have the initial time to compare. This observation also applies to results presented in item 3.3, figs 4 and 5.
As stated by the authors in ln 44: “ development of particular toxic and/or non-toxic subpopulations within cyanobacteria is impossible to predict. Also, among P. agardhii populations, both MC-producing and non-MC-producing strains occur, and the genotypic variation concerns both MCs and other biologically active oligopeptides...the populations of P. agardhii are not clonal and may be composed of several chemotypes.”
Response: We improved the description of the experimental design and clarified that at the beginning of the experiments the compositions of P. aghardii population in each treatment was uniform as we used one and the same water sample that was mixed up for homogeneity before the experiments set up (lines 128-130).
- No dose- effect relationship was apparent in fig 3.
Response: The information was completed. Lines 318-319. “In the presence of both extracts, content of MCs in P. agardhii biomass increased, however, dose-effect relationship was not strong”
- Also, we do not know about “production “, abundance is a better term.
Response: “Production” was changed to “content” (line 318)
- Table 4 similarity of the composition of oligopeptides found in the cyanobacterial extracts, control P. agardhii and P. agardhii exposed to the extracts
A hetamap would be more informative, also with data from fig 4. The replicates should be presented individually.
Response: We ask to leave the result in a present form.
- Ln 299-323 If data are presented in a figure or table it is unnecessary to describe in the text.
Response: lines 385-392. We are ask to leave the description as it explains and compare the results presented in different Tables and Figure. Some details were deleted, however.
Discussion
- Ln 340-352 This paragraph concentrates much of my concerns about this study.
“ in the presence of cyanobacterial extracts, obtained from two different cyanobacterial scums formed by P. agardhii and containing a high number and diverse composition of bioactive oligopeptides (MCs and other mostly non-ribosomal peptides) and high concentrations of biogenic compounds (nitrogen and phosphorus) the growth of another P. agardhii population and its content of Chl-a and MCs increased. “
There is no convincing evidence showing that the increase in P. agardhii biomass was not only the consequence of the biogenic compounds (nitrogen and phosphorus). As mentioned by the authors (ln 359). Control condition poorly described.
Response: (lines 402-406) The sentence was changed to: Our study showed for the first time that in the presence of a high number and diverse composition of bioactive oligopeptides (MCs and other mostly non-ribosomal peptides) and high concentrations of biogenic compounds (nitrogen and phosphorus) the final accumulation of P. agardhii biomass and its content of Chl-a and MCs increased.
- There is no convincing evidence that content of Chl-a and MCs increased per cell.
Response: The content of Chl-a and MCs increased per mg of P. agardhii biomass.
- There is no data on growth, only the final accumulation of P. agardhii biomass.
Response: lines 405-406. Growth was changed to “final accumulation of P. agardhii biomass”.
- “Changes in the peptide profiles of the P. agardhii community were observed after exposure to two different extracts. “
As mentioned before, it could not be distinguished from temporal fluctuation in the population incubated in laboratory conditions. No data corresponding to the initial time is presented.
Response: The changes were confirmed by comparison with the control analysed after exposure (Table S1, Figure 5)
- “ As we exposed a natural population of the toxigenic P. agardhii to the extracts of natural cyanobacterial biomasses predominated (82 and 95%) by this species and containing a little admixture (5 and 18%) of other cyanobacteria, we could observe more complex phenomena, resulting both from intra- and interspecific interactions, than in the case of experiments on individual laboratory strains. “
The experimental design combines two complex materials, a natural population of P. agardhii and extracts of natural cyanobacterial communities containing P. agardhii and other cyanobacteria. Thus, it is extremely difficult to establish a causal relationship between extract composition and physiological response. Although the title refers to intraspecific interactions, this issue is not covered here using extracts.
Response: lines 409-413. The title was changed to more direct. The sentence was changed to; “As we studied a natural population of the toxigenic P. agardhii, we could observe more complex phenomena than in the case of experiments on laboratory strains.”
- ln 363 “The high abundance of oligopeptides coming from the scum extracts can have served as valuable substrates for cyanobacterial growth, especially, when nutrients become more limiting after a few days of exposure. “
Nutrient limitation was not shown in the results. The proportion of oligopeptides to total protein or N containing molecules is so small that they would not supply nutrient deficiency.
Response: lines 427-428. The confusing part of the sentence was removed. “The high abundance of oligopeptides coming from the scum extracts can have served as valuable substrates for cyanobacteria.”
- Ln 374 “Our experiments showed that the content of Chl-a in the exposed P. agardhii correlated with increasing extract concentrations independently from broad MC concentration.”
No correlation was clearly shown in the results.
Response: lines 348-349. The sentence was rewritten to “Our results showed that the content of Chl-a in the exposed P. agardhii in both experiments changed in a similar way.”
- Ln 376 ”Moreover, the Planktothrix biomass plateaued while the Chl-a content kept increasing with the increased MC concentration.”
Also, not clear from results. And, as stated by the authors below, this can be the effect of turbidity, that was not monitored.
Response: Statistically significant results are presented on figs 1 and 2. Turbidity was not monitored in these microcosm experiments but it was discussed (for example line 474 ).
- Ln 379 “Moreover, the Planktothrix biomass plateaued while the Chl-a content kept increasing with the increased MC concentration.”
Here is another possible limitation argued by the authors, the resulting culture conditions after the addition of the extracts to P. aghardii cultures was not detailed, as mentioned before. It is expected that the extracts resulted in higher turbidity in comparison to control, thus another variable was introduced in the experiment. This is mentioned again in ln 413.
Response: We agree with this statement.
- ln 381 “Our results strongly suggest that dissolved P. agardhii metabolites and other cell compounds containing biogenic substances allow efficient growing and, therefore, domination of P. agardhii in natural cyanobacterial assemblages when water blooms caused by other species collapse.”
I strongly disagree. The effect can be due to nutrient input, this would happen in the collapse of any cyanobacterial bloom.
Response: lines 441-443. The sentence was rewritten “Our results suggest that cyanobacterial metabolites containing biogenic substances may allow efficient development of P. agardhii in natural cyanobacterial assemblages when water blooms caused by other species collapse [11] with a release of cell content”.
- ln 385 “Future experiments with ... similar amounts of available N, P and C-substrate can confirm our findings.”
In fact, this should have been your control in this study.
Response: We agree.
- item 4.2 there is a confusion in the text to explain a possible effect of extracts on MC amounts. The term “production” is misleading because there was no experiment to measure this, only MC amount. Also, MC quotas were not determined. As the study deals with a natural population, it is a mixture of MC+ and MC- strains, and their contribution is unknown, and can change with the incubation period. So, it is extremely difficult to discuss the effect of the extracts on MC production. Finally, MC content will increase simply as a result of biomass increase.
Response: The term production was changed (line 450). Mc content is presented in our work per P. agardhii biomass (per mg) not per liter of water so it was not connected with increase in biomass.
- Ln 423 “dissolved oligopeptides present in the extracts did not accumulate in P. agardhii under exposure conditions “
It is surprising to find this discussion since this was not presented as an objective of this study and experiments were not designed to test this.
Response: The aim of this work was to study the effect of the extracts on P. agardhii, incuding effect on oligopeptides content (lines 19-22). The similarity index used in this study (Tabel 4) showed low similarity between oligopeptides profiles in P. agardhii exposed and extracts and high between P. agardhii exposed to both extracts. This suggest lack accumulation. However, the sentence was improved and the title of the paragraph was changed to “Changes in oligopeptides profiles and single compounds in P. agardhii exposed.” Line 486.
- Ln 441 “Our experiment showed that the changes in oligopeptide composition, suggesting changes in abundance of particular chemotypes in the P. agardhii population, may be quick as a response to dissolved compounds of other chemotypes of this species or other cyanobacterial species”
There is no evidence in the results to support the statement that changes in abundance of particular chemotypes in the P. agardhii population (that were not characterized) may be a response to dissolved compounds of other chemotypes (that were not characterized either), and in this study these compounds were a minor component of a rich nutritional and complex extract.
Response: This was a suggestion discussed with other papers and the sentence was rewritten (lines 513-515): Our experiment may suggest that the changes in oligopeptide composition, suggesting changes in abundance of particular chemotypes in the P. agardhii population, may be quick as a response to dissolved compounds (including nutrients) of other cyanobacteria.
- Ln 447 “our findings suggest that in natural conditions cyanobacterial metabolites that are released to the water during bloom formation or collapse, may play some role in the regulation of cyanobacterial development at the chemotype level.”
Same observation as above.
Response: The sentence was deleted.
- Although all the discussion is highly speculative and does not relate to the correct interpretation of the results presented, item 4.4 is specially so.
Response: The speculative parts part were deleted. The paragraph was shortened and leaved sentences were connected with the previous paragraph.
Reviewer 2 Report
The clarity of the English can be improved, from minor uses of singular and plural, to the clarity of some sentences.
See line 233 which has an obscure meaning
line title 422'exposed' needs more information
conclusion lines 489 to 493. This important sentence needs re-writing so that its meaning is clear. it can best be expressed as two sentences, as it is too long.
The discussion is very detailed, and may be best shortened to the main points.
The field is an interesting one, but it is unclear to me whether the effects are due to the nutrient effects of the extracts, or to organic materials such as the peptides. The impression that I get from reading the paper is that the nutrients effects are predominant. perhaps this can be included in the conclusions.
Author Response
Response to the comments of the Reviewer 2
Thank you for valuable suggestions.
1: The clarity of the English can be improved, from minor uses of singular and plural, to the clarity of some sentences.
See line 233 which has an obscure meaning
Response: The clarity of the English has been improved, also in line 233 (line 278 now) “After the 7-day exposure to two extracts, at the first three increasing extract’s concentrations, the P. agardhii biomass increased several times in comparison with the controls (Figure 2).”
2: line title 422'exposed' needs more information
Response: The information has been added (line 492) “Changes in oligopeptides profiles and single compounds in P. agardhii exposed to the extracts”
- conclusion lines 489 to 493. This important sentence needs re-writing so that its meaning is clear. it can best be expressed as two sentences, as it is too long.
Response: The sentence was rewritten (lines 564). As the similarity between the oligopeptide profiles was the highest for the P. agardhii exposed to both extracts that differed considerably in oligopeptide composition but contained high concentrations of nutrients, it seems that biogenic compounds, not oligopeptides themselves, positively influenced on a mixed natural P. agardhii population
- The discussion is very detailed, and may be best shortened to the main points.
Response: The discussion was rewritten and shortened. For example, lines 403, 410-413, 452-453, 476-479, 519-521, 527-532,534-536, 539-543, 552-556 were deleted.
- The field is an interesting one, but it is unclear to me whether the effects are due to the nutrient effects of the extracts, or to organic materials such as the peptides. The impression that I get from reading the paper is that the nutrients effects are predominant. perhaps this can be included in the conclusions.
Response: The results and their description were were clarified. Nutrients effect is predominant and it was highlighted in conclusions (Lines 566-567).
Reviewer 3 Report
Comments to the Manuscript: ijerph-754692
The manuscript describes a microcosmos experiment of the effect of P. agardhii bloom samples extracts in the growth and peptide production of other P. agardhii bloom sample. Its major findings were growth stimulation of P. agardhii and differential microcystin production in the microcosmos sample. The study addresses an important and current issue of cyanopeptides and toxins in the environment and their effects with direct consequences for human health. The research is presented with care and in detail. However, some issues need clarification. It is not clear the effect on chlorophyll production and its relation to the extracts. Microcosmus experiments and whole samples experiments are tricky since many organisms are present that interact with each other and respond differently. Organisms will respond to the simple fact of artificial growth conditions. It is also not clear if culture medium was added to the bloom sample, depending on the culture medium used there will be also a selection of the organisms. A graphical scheme with the experimental design will greatly improve the manuscript, with information relating: volume of the experiment, controls, replicates, volume of extract for volume of bloom sample, aeration, culture medium, experimental samples volume taken, parameters measured in each sample, sample preparation (filtration/ whole sample). Furthermore, a control of the extracts by them selves was not executed to verify compound degradation during experiment conditions and was not discussed also. Also, the possible change in microbiota with the extract application in the bloom sample was not study or discussed. The discussion is speculative in general and lacks focus.
In sum, I would strongly recommend rewriting the manuscript specially the results and the discussion, make it simpler and focus to the amount of information that can be better described and discussed avoiding speculation without evidence of data.
For the reasons mentioned above, the manuscript is worth for being published on the Journal after major revisions.Specific comments:
Introduction:
Line 35-38: The paragraph is confusing please rephrase (Example: How “less known compounds” make “more evident” the “harmful effects on the health of humans and on the functioning of aquatic biocenoses”?)
Line 40: The affirmation “…may intensify water blooms caused by this species” is speculative by itself. It should be developed and supported by other studies.
Line 64 – 66: Please clarify and explain further: “For example, in bloom-forming cyanobacteria P. agardhii and P. rubescens a high frequency of genes involved in synthesis of several toxins and other bioactive peptide were detected, but not in other Planktothrix species” and support the sentence with bibliographic references.
Materials and methods
Line 84 – 115: A schematic illustration of the experimental design will greatly improve the manuscript in the understanding of how the experiments were executed.
Line 107: Please explain in the text the concept of “biologically active MCs”
Results:
Line 224: In table 2 the number of pepetides in the experimental samples (P. agardhii exposed to extract Pa-A and P. agardhii exposed to extract Pa-B) is inferior to the original
extracts (Pa-A and Pa-B). Could you explain that, was there any evidence of degradation of the peptides during the experiments? Please discuss.
Line 260: Figure 2 and Table 3 represent the same results, since contribution of P. agardhii for the biomass is above 97% it will be expected that chl-a in P. agardhii biomass would be almost the same as total Chl-a. Please clarify the results presented and Table 3 could be supplementary material.
- The Tables format should be improved and uniformized
- Figures resolution must be improved.
Discussion:
Line 341 – 344 and Line 348 - 352: These paragraphs are poorly constructed and confusing please rephrase them.
Line 466: M. aeruginosa should be in italic please check throughout the text.
Author Response
Response to the comments of Reviewer 3
- The manuscript describes a microcosmos experiment of the effect of P. agardhii bloom samples extracts in the growth and peptide production of other P. agardhii bloom sample. Its major findings were growth stimulation of P. agardhii and differential microcystin production in the microcosmos sample. The study addresses an important and current issue of cyanopeptides and toxins in the environment and their effects with direct consequences for human health. The research is presented with care and in detail.
Response: Thank you very much for this opinion.
- However, some issues need clarification. It is not clear the effect on chlorophyll production and its relation to the extracts. Microcosmus experiments and whole samples experiments are tricky since many organisms are present that interact with each other and respond differently. Organisms will respond to the simple fact of artificial growth conditions. It is also not clear if culture medium was added to the bloom sample, depending on the culture medium used there will be also a selection of the organisms.
Response: We clarified the unclear issues and methods, results and discussion have been re-written and improved. Scheme presenting the experimental design and conditions was added (Figure 1).
- A graphical scheme with the experimental design will greatly improve the manuscript, with information relating: volume of the experiment, controls, replicates, volume of extract for volume of bloom sample, aeration, culture medium, experimental samples volume taken, parameters measured in each sample, sample preparation (filtration/ whole sample).
Response: The graphical scheme with suggested information was added (Figure 1)
- Furthermore, a control of the extracts by them selves was not executed to verify compound degradation during experiment conditions and was not discussed also. Also, the possible change in microbiota with the extract application in the bloom sample was not study or discussed. The discussion is speculative in general and lacks focus.
Response: The discussion was rewritten and shortened. Control conditions are better described and are presented in Table 1. We did not add any nutrients in control variants of experiments. The controls used were just lake water from Syczynskie Lake sampled in 2015, left without the addition of any substances. The aim of this work was to study the effects of mixtures of cyanobacterial metabolites on a natural population of P. agardhii.
- In sum, I would strongly recommend rewriting the manuscript specially the results and the discussion, make it simpler and focus to the amount of information that can be better described and discussed avoiding speculation without evidence of data.
Response: The manuscript (specially the results and discussion) was rewritten according to the suggestions.
- For the reasons mentioned above, the manuscript is worth for being published on the Journal after major revisions.
Response: Thank you very much for this opinion.
Specific comments:
Introduction:
- Line 35-38: The paragraph is confusing please rephrase (Example: How “less known compounds” make “more evident” the “harmful effects on the health of humans and on the functioning of aquatic biocenoses”?)
Response: The paragraph was rewritten (lines 48-51). There is: “Cyanobacterial extracts can harm humans and affect the functioning of aquatic biocenoses [2,4,5] particularly when the blooms are formed by strains producing cyanotoxins such as hepatotoxic microcystins and cylindrospermopsins, neurotoxic anatoxin-a, anatoxin-a(S), saxitoxins or other less known compounds [6-9]. “
- Line 40: The affirmation “…may intensify water blooms caused by this species” is speculative by itself. It should be developed and supported by other studies.
Response: The sentence was developed and supported by literature. There is: “Ongoing worldwide eutrophication problems [13] may intensify water blooms caused by this species in eutrophic freshwaters. For example, Doculil and Teubner [14] showed that the biomass of P. agardhii increased with increasing concentrations of total phosphorus.
- Line 64 – 66: Please clarify and explain further: “For example, in bloom-forming cyanobacteria P. agardhii and P. rubescens a high frequency of genes involved in synthesis of several toxins and other bioactive peptide were detected, but not in other Planktothrix species” and support the sentence with bibliographic references.
Response: (lines 80-83).The sentence was clarified and the reference was added. “For example, a high frequency of genes involved in synthesis of several toxins and other bioactive peptides were detected in bloom-forming cyanobacteria P. agardhii and P. rubescens, but not in other non-bloom-forming Planktothrix species [12].”
Materials and methods
- Line 84 – 115: A schematic illustration of the experimental design will greatly improve the manuscript in the understanding of how the experiments were executed.
Response: The schematic illustration was added (Figure 1). The paragraph describing experiments was improved (lines 123-153).
- Line 107: Please explain in the text the concept of “biologically active MCs”
Response: The confusing term was deleted.
Results:
- Line 224: In table 2 the number of pepetides in the experimental samples (P. agardhii exposed to extract Pa-A and P. agardhii exposed to extract Pa-B) is inferior to the original extracts (Pa-A and Pa-B). Could you explain that, was there any evidence of degradation of the peptides during the experiments? Please discuss.
Response: The title of the table has been refined. The inaccuracy has been cleared up. The table contains information about number of oligopeptides found in two different extracts (Pa-A and Pa-B) of P. agardhii-dominated scum samples, and in the biomass of the control P. agardhii and P. agardhii after exposure to the highest concentrations of the extracts (line 269). We tested the degradation of MCs during the experiments (lines 152-154, there is: “Dissolved MCs were analysed in filtered (GF/C Whatman filters) water obtained from the variants with the highest concentrations of the extracts to control their stability during the experiment.”) and the results were described (lines 292-295; there is “Analysis of dissolved MCs in the water from both experiments with the highest concentrations of the extracts Pa-A and Pa-B (treatment IV) showed that after 7 days the MC content decreased by 50 and 59% in variants with the extracts Pa-A and Pa-B, respectively (in comparison to their initial concentrations).”)
- Line 260: Figure 2 and Table 3 represent the same results, since contribution of P. agardhii for the biomass is above 97% it will be expected that chl-a in P. agardhii biomass would be almost the same as total Chl-a. Please clarify the results presented and Table 3 could be supplementary material.
Response: We ask to leave Figure 2 (now Figure 3) and Table 3 in their present form and place as they show different results. Figure 3 presents Chl-a content in P. agardhii biomass (in µg per milligram of P. agardhii) after the 7-day exposure to the extracts Pa-A and Pa-B. The Chl-a content was variable depending on the concentration of extracts. Table 2 shows percentage contribution of the biomass of particular cyanobacterial taxa in the total biomass of phytoplankton after the 7-day exposure to extracts and in the case of P. agardhii, it was stable. The title of the Figure 2 was improved.
- The Tables format should be improved and uniformized
Response: The Tables format was improved and uniformized.
- Figures resolution must be improved.
Response: Figure resolution was improved. Original figures with resolution required by the journal were also uploaded in the submission system.
Discussion:
- Line 341 – 344 and Line 348 - 352: These paragraphs are poorly constructed and confusing please rephrase them.
Response: These sentences and the whole introduction to Result were deleted according to the suggestion of the Reviewer 1.
- Line 466: aeruginosa should be in italic please check throughout the text.
Response: Changed throughout the text.
Round 2
Reviewer 1 Report
The authors reformulated the description of the results and thus presented a more focused, simple and consistent study. Also, the discussion is more concrete and focused on the results. The methodology description was improved with the inclusion of additional information, as recommended. Finally, the study has a clear hypothesis. Still, some suggestions are listed below in an attempt to further improve data presentation and clarity.
Introduction
Introduction explores mostly the effect of oligopeptides or bioactive compounds, when in fact the tested and observed effect is strongly associated with the nutritional component. Therefore, the introduction should mention the possible effect of bloom senescence on natural phytoplankton communities considering the complex situation simulated with the tested extracts.
39
Cyanobacterial extracts can harm humans
- Cyanobacterial blooms can harm humans
Methods
175
After collection and phytoplankton enumeration (described in point 2.3), the scum samples were frozen (at -20°C) until the day of extraction, analyses and experiments.
- How did scum samples became extracts, just freeze and thaw? Please specify
Fig 1
four flasks thus I II II IV correspond to extract dilutions and this is not clear
table in this fig “final” concentrations of extracts substances? This is not final because the next arrows correspond to 7 days of exposure and “ after exposure” and this is final. In the table should be “initial concentrations” in the extracts or composition of extract dilutions ?
244
both cyanobacterial scum samples were analysed for taxonomic composition and cyanobacterial enumeration in a plankton Sedgewick-Rafter chamber under a light microscope. Cell biovolume was measured to determine cyanobacterial biomass [35]. It was assumed that the specific weight of planktonic microalgae is 1.0 ug ml1 [36] and, therefore, the biomass was expressed in mg l -1 of water.
- The sample was first analyzed for cell enumeration. Number of cells was converted to cell biovolume (how? A conversion factor only for P aghardii, for all cyanobacteria, an average value?), biovolume was converted to biomass (how? A general conversion factor all cyanobacteria?) and results are shown as biomass per volume. With so many conversions, each step should be described.
This was used to all cyanobacetria in general? You were interested in P. aghardii biomass (major biomass in your samples), did you use a conversion specific for this species?
Table 1
Biomasses, extracts and sample used in experiments (as controls and treated)
change for “samples”
decimal numbers with (.) Or (,) or nothing → correct
extract code - delete
497
rephrase
The extracts were rich in biogenic compounds (Table 1) that were added in different dilutions in the experiments (Figure 1).
506
For the first time it has been documented that cyanobacterial extracts obtained from two P. agardhii dominated scum samples
- “For the first time it has been documented that ” delete
510
rephrase
- and MCs content (Figure 4) in another P. agardhii dominated natural sample
511
After the 7-day exposure to two extracts, at the first three increasing extract’s concentrations, the P. agardhii biomass increased several times in comparison with the controls (Figure 2).
Fig 2
Different lowercase letters (a-b) indicate statistically significant differences in P. agardhii biomass
Thus, according to your statistical analysis all tested concentrations led to a similar increase in P. agardhii biomass in comparison to controls. Please rephrase ln 511.
514
At the highest concentration of the extract Pa-A, the biomass slightly decreased, whereas, at the highest concentration of the extract Pa-B, it stopped growing in comparison with the preceding treatment
Same comment above
517
At the end of exposure, the filaments of P. agardhii exposed to the extract Pa-B were longer (mean length equal to 280 µm) than the filaments of the cyanobacterium exposed to the extract Pa-A (250 µm).
Data not shown? We do not know if their size were already different at the begining of the test. Is this information relevant? No context, consider deleting.
Figure 2.
caption, please rephrase
- The biomass of P. agardhii after the 7-day exposure to the extracts Pa-A (A) and Pa-B (B) originally containing different concentrations of MCs. Both extracts dilutions had similar concentration of Chl-a
Table 3.
Contribution of the biomass (%) of particular cyanobacterial taxa in the total biomass of phytoplankton after the 7-day exposure to extracts Pa-A and Pa-B of different concentration of MCs and similar concentration of Chl-a
rephrase as suggested for fig 2 caption
The dilutions of the extracts in the table should be represented as Chl concentratios in accordance with your methods:
191
The experiments were performed for seven days. Four dilutions of the extracts of the similar final concentration range of extracellular Chl-a (as an equivalent of cyanobacterial biomass) in water, i.e. 0.33–2.58 mg l−1 in the treatments with the extract Pa-A and 0.33–2.63 mg l−1 in the treatments with the extract Pa-B, were used.
Table 3 can be presented as supplementary
528
The biomass even slightly increased after exposure to higher concentrations of the extract Pa-B, than to PaA
Was this difference significant? If not, delete. Also, the extracts differ in other metabolites not only MC.
Figure 3.
caption, change as cited for fig 2 and delete bioactive
587
the contribution of both analogues to the total MC content changed with changing extract concentrations (Figure 3) → Fig 4
In both experiments, the highest contribution of dmMC-LR was observed in the conditions when cyanobacterial biomass was equal to 1.3 mg Chl-a l−1
difficult to follow because data (contribution of dmMC-LR and cyanobacterial biomass) are in two different figures
Maybe include a summary fig with control x treatment (with more concentrated extract, 2.5 mg/L Chl), and all the parameters together, biomass, Chl, MC
651-680
Figure 4 is now Figure 5, correct
Figure 5 and Figure 6
the data can be represented in one figue with all peptides and their relative amount in the 3 profiles: control, PaA an PaB
It is not clear why 2 figures were presented since fig 5 also contains oligopeptide classes other than MCs
Figure 5. Relative content of particular oligopeptides in P. agardhii exposed to the extracts obtained from two P. agardhi-dominated scum samples
Figure 6. The composition of particular oligopeptide classes other than MCs in the biomass of P. agardhii exposed to the extracts Pa-A and Pa-B.
Discussion
713
Our study showed for the first time that in the presence of a high number and diverse composition of bioactive oligopeptides (MCs and other mostly non-ribosomal peptides) and high concentrations of biogenic compounds (nitrogen and phosphorus) the final accumulation of P. agardhii biomass and its content of Chl-a and MCs increased
- You could rephrase and make a parallel with the senescence of a bloom when this situation would occur.
The addition of nutrients led to an increase in P aghardii biomass and this is not so surprising, thus remove “for the first time”
815
Nevertheless, MCs themselves may also exert some positive effects on cyanobacteria. For example, MC‐RR exposure caused a significant increase in the production of extracellular polysaccharides (EPS) in Microcystis, although it did not influence cyanobacterial growth rate [49]
Cellular aggregation is commonly described as a stress response in Microcystis, not necessarily a positive effect. Please explain why it is cited as positive here.
820
There are some reports indicating that bioactive cyanobacterial compounds themselves are not essential for cyanobacterial growth [51,52]
This sentence does not fit here. No relation to the rest of the paragraph. Remove or complete the idea.
992
in which P. agardhii biomass decreased or stopped growing, may suggest that dmMC-LR might play some role in the regulation of P. agardhii biomass.
Suggestion: may suggest that dmMC-LR content might be linked to P. agardhii cell density.
997
4.3. Changes in oligopeptides profiles and single compounds in P. agardhii exposed to the extracts
Suggestion : Changes in oligopeptide profiles in P. agardhii populations exposed to the extracts
Conclusions
1221
The obtained results showed that when lysed, P. agardhii cells release into the water compounds that may increase the biomass as well as the content of Chl-a and MCs in a natural P. agardhii population.
1222
MCs present in the extracts did not affect the increase in P. agardhii biomass.
1223
Although increasing concentrations of P. agardhii extracts did alter total MC content, changes in the relative contributions of dmMC-LR and MC-RR derivatives were found.
1226
The similarity between the oligopeptide profiles was highest for P. agardhii exposed to the each extract. As the two tested extracts differed considerably in oligopeptide composition and contained similar high concentrations of nutrients, it seems that biogenic compounds, not oligopeptides themselves, positively influenced the biomass accumulation of a natural P. agardhii population
Author Response
Response to the comments of the Reviewer 1.
Thank you very much for all valuable comments and suggestions. All were taken into consideration.
- The authors reformulated the description of the results and thus presented a more focused, simple and consistent study. Also, the discussion is more concrete and focused on the results. The methodology description was improved with the inclusion of additional information, as recommended. Finally, the study has a clear hypothesis. Still, some suggestions are listed below in an attempt to further improve data presentation and clarity.
Response: Thank you very much for the opinion.
Introduction
- Introduction explores mostly the effect of oligopeptides or bioactive compounds, when in fact the tested and observed effect is strongly associated with the nutritional component. Therefore, the introduction should mention the possible effect of bloom senescence on natural phytoplankton communities considering the complex situation simulated with the tested extracts.
Response: lines 89-95. The introduction was improved to mention the possible effect of bloom senescence on natural phytoplankton communities considering the complex situation simulated with the tested extracts. “In addition to peptides, which may be present in large numbers and quantities in cyanobacterial biomass [17,18,21,24], the presence of other components, such as biogenic compounds or minerals may influence phytoplankton communities. For example, Suikkanen et al. [33] showed that cyanobacterial filtrates containing unknown compounds stimulated both colonial (Snowella spp.) and filamentous cyanobacteria (Pseudanabaena spp., Anabaena spp., Aphanizomenon sp., N. spumigena), a chlorophyte (Oocystis sp.), a dinoflagellate (Amphidinium sp.) and nanoflagellates, but inhibited cryptophytes.”
- 39 Cyanobacterial extracts can harm humans - Cyanobacterial blooms can harm humans
Response: line 50: “Cyanobacterial extracts can harm humans” was changed to “Cyanobacterial blooms can harm humans”
Methods
- 175 After collection and phytoplankton enumeration (described in point 2.3), the scum samples were frozen (at -20°C) until the day of extraction, analyses and experiments.
- How did scum samples became extracts, just freeze and thaw? Please specify
Response: line 126. We added the information that “The extraction procedure is described in section 2.4.” The extraction was described in more detail in lines 188-191: “Dense bloom samples of P. agardhii-dominated biomasses were used to prepare crude extracts for the determination of MCs, and for experiments. The samples were sonicated (for 20 min, 50 W; SONOPULS ultrasonic homogeniser, Bandelin) and after centrifugation (14,000 rpm for 10 min at 17°C) supernatants were collected. Cyanotoxins were analysed in the extracts before experiments.”
- Fig 1
four flasks thus I II II IV correspond to extract dilutions and this is not clear
table in this fig “final” concentrations of extracts substances? This is not final because the next arrows correspond to 7 days of exposure and “ after exposure” and this is final. In the table should be “initial concentrations” in the extracts or composition of extract dilutions ?
Response: Improved: “Treatments (triplicates)” was changed to “Extract dilutions (each in triplicate)”. Table: “final” was changed to “initial”.
- 244 both cyanobacterial scum samples were analysed for taxonomic composition and cyanobacterial enumeration in a plankton Sedgewick-Rafter chamber under a light microscope. Cell biovolume was measured to determine cyanobacterial biomass [35]. It was assumed that the specific weight of planktonic microalgae is 1.0 ug ml1 [36] and, therefore, the biomass was expressed in mg l -1 of water.
- The sample was first analyzed for cell enumeration. Number of cells was converted to cell biovolume (how? A conversion factor only for P aghardii, for all cyanobacteria, an average value?), biovolume was converted to biomass (how? A general conversion factor all cyanobacteria?) and results are shown as biomass per volume. With so many conversions, each step should be described.
This was used to all cyanobacetria in general? You were interested in P. aghardii biomass (major biomass in your samples), did you use a conversion specific for this species?
Response: The description was improved. Lines 170-174. “Cyanobacterial filaments or colonies were counted. From 30 to 50 filaments or colonies were measured to estimate the cell biovolume [35] and to determine cyanobacterial biomass by multiplying the number of filaments or colonies by their biovolume. It was assumed that the specific weight of planktonic microalgae is 1.0 g ml-1 [36] and, therefore, the biomass was expressed in mg l-1 of water.”
- Table 1
Biomasses, extracts and sample used in experiments (as controls and treated)
change for “samples”
decimal numbers with (.) Or (,) or nothing → correct
extract code - delete
Response: “sample” was changed to “samples”. Decimal numbers were corrected. “Extract code” was deleted.
- 497 rephrase
The extracts were rich in biogenic compounds (Table 1) that were added in different dilutions in the experiments (Figure 1).
Response: lines 277-278. The sentence was rephrased according to the suggestion: “The extracts were rich in biogenic compounds (Table 1) that were added in different dilutions in the experiments (Figure 1).”
- 506
For the first time it has been documented that cyanobacterial extracts obtained from two P. agardhii dominated scum samples
- “For the first time it has been documented that ” delete
Response: line 286 “For the first time it has been documented that ” was deleted.
- 510 rephrase
- and MCs content (Figure 4) in another P. agardhii dominated natural sample
Response: Line 289-290: The sentence was changed to “and MC content (Figure 4) in another P. agardhii dominated natural sample.”
- 511
After the 7-day exposure to two extracts, at the first three increasing extract’s concentrations, the P. agardhii biomass increased several times in comparison with the controls (Figure 2).
Fig 2 Different lowercase letters (a-b) indicate statistically significant differences in P. agardhii biomass
Thus, according to your statistical analysis all tested concentrations led to a similar increase in P. agardhii biomass in comparison to controls. Please rephrase ln 511.
Response: lines 290-293. The sentence was rewritten “After the 7-day exposure to two extracts, the P. agardhii biomass increased several times in comparison with the controls, however, the differences between particular treatments were not statistically significant (Figure 2).”
- 514
At the highest concentration of the extract Pa-A, the biomass slightly decreased, whereas, at the highest concentration of the extract Pa-B, it stopped growing in comparison with the preceding treatment
Same comment above
Response: The sentence was deleted as the corrected sentence in lines 290-293 comments the results.
- 517
At the end of exposure, the filaments of P. agardhii exposed to the extract Pa-B were longer (mean length equal to 280 µm) than the filaments of the cyanobacterium exposed to the extract Pa-A (250 µm).
Data not shown? We do not know if their size were already different at the begining of the test. Is this information relevant? No context, consider deleting.
Response: We agree. The sentence was deleted (lines 298-300).
- Figure 2.
caption, please rephrase
- The biomass of P. agardhii after the 7-day exposure to the extracts Pa-A (A) and Pa-B (B) originally containing different concentrations of MCs. Both extracts dilutions had similar concentration of Chl-a
Response: lines 321-323. The caption was rephrased: “The biomass of P. agardhii after the 7-day exposure to the extracts Pa-A (A) and Pa-B (B) originally containing different concentrations of MCs. Both extracts dilutions had similar concentration of Chl-a.”
- Table 3.
Contribution of the biomass (%) of particular cyanobacterial taxa in the total biomass of phytoplankton after the 7-day exposure to extracts Pa-A and Pa-B of different concentration of MCs and similar concentration of Chl-a
rephrase as suggested for fig 2 caption
The dilutions of the extracts in the table should be represented as Chl concentratios in accordance with your methods:
191 The experiments were performed for seven days. Four dilutions of the extracts of the similar final concentration range of extracellular Chl-a (as an equivalent of cyanobacterial biomass) in water, i.e. 0.33–2.58 mg l−1 in the treatments with the extract Pa-A and 0.33–2.63 mg l−1 in the treatments with the extract Pa-B, were used.
Table 3 can be presented as supplementary
Response: The title of the table was rephrased: “Contribution of the biomass (%) of particular cyanobacterial taxa in the total biomass of phytoplankton after the 7-day exposure to extracts Pa-A and Pa-B originally containing different concentrations of MCs. Both extracts dilutions had similar concentration of Chl-a. Data are expressed as means ± SE, n=3.”
The dilutions of the extracts are presented as Chl-a concentrations. Table 3 is presented as supplementary Table S2 so it was deleted from the text.
- 528
The biomass even slightly increased after exposure to higher concentrations of the extract Pa-B, than to PaA
Was this difference significant? If not, delete. Also, the extracts differ in other metabolites not only MC.
Response: lines 310-312. The sentence was deleted.
- Figure 3.
caption, change as cited for fig 2 and delete bioactive
Response: lines 328-330. The caption was changed, the “bioactive” was deleted. “Chl-a content in P. agardhii biomass after the 7-day exposure to the extracts Pa-A (A) and Pa-B (B) originally containing different concentrations of MCs. Both extracts dilutions had similar concentration of Chl-a.”
- 587
the contribution of both analogues to the total MC content changed with changing extract concentrations (Figure 3) → Fig 4
In both experiments, the highest contribution of dmMC-LR was observed in the conditions when cyanobacterial biomass was equal to 1.3 mg Chl-a l−1
difficult to follow because data (contribution of dmMC-LR and cyanobacterial biomass) are in two different figures
Maybe include a summary fig with control x treatment (with more concentrated extract, 2.5 mg/L Chl), and all the parameters together, biomass, Chl, MC
Response: lines 344-347. The sentence applies to the data presented only in Figure 4 and was rebuilt so that it is not confusing now. Now there is: “In both experiments, the highest contribution of dmMC-LR was observed in the conditions with extracts dilutions containing 1.3 mg Chl-a l−1. dmMC-LR reached 21% and 29% of the total MC content in P. agardhii exposed to the extract Pa-A and Pa-B, respectively.”
- 651-680
Figure 4 is now Figure 5, correct
Response: line 373. Corrected.
- Figure 5 and Figure 6
the data can be represented in one figure with all peptides and their relative amount in the 3 profiles: control, PaA an PaB
It is not clear why 2 figures were presented since fig 5 also contains oligopeptide classes other than MCs
Figure 5. Relative content of particular oligopeptides in P. agardhii exposed to the extracts obtained from two P. agardhi-dominated scum samples
Figure 6. The composition of particular oligopeptide classes other than MCs in the biomass of P. agardhii exposed to the extracts Pa-A and Pa-B.
Response: The data are presented in one figure 5 according to the suggestion.
Discussion
- 713
Our study showed for the first time that in the presence of a high number and diverse composition of bioactive oligopeptides (MCs and other mostly non-ribosomal peptides) and high concentrations of biogenic compounds (nitrogen and phosphorus) the final accumulation of P. agardhii biomass and its content of Chl-a and MCs increased
- You could rephrase and make a parallel with the senescence of a bloom when this situation would occur.
The addition of nutrients led to an increase in P aghardii biomass and this is not so surprising, thus remove “for the first time”
Response: line 419. “for the first time” was deleted.
- 815
Nevertheless, MCs themselves may also exert some positive effects on cyanobacteria. For example, MC‐RR exposure caused a significant increase in the production of extracellular polysaccharides (EPS) in Microcystis, although it did not influence cyanobacterial growth rate [49]
Cellular aggregation is commonly described as a stress response in Microcystis, not necessarily a positive effect. Please explain why it is cited as positive here.
Response: The authors of the cited paper write that “Cellular release of MCs, therefore, may play a key role in the persistence of algal colonies and the dominance of Microcystis.” This suggests positive effect. The cited sentence was added.
This part was rewritten (lines 451-457): “Nevertheless, MCs themselves may also exert some positive effects on cyanobacteria colony size. For example, MC‐RR exposure caused a significant increase in the production of extracellular polysaccharides (EPS) in Microcystis, although it did not influence cyanobacterial growth rate [49]. Moreover, both MC-RR and MC-LR (at concentrations 0.25–10 µg l−1) significantly enhanced Microcystis colony sizes. Therefore, according to Gan et al. [49] cellular release of MCs may play a key role in the persistence of cyanobacterial colonies and the dominance of Microcystis.”
- 820
There are some reports indicating that bioactive cyanobacterial compounds themselves are not essential for cyanobacterial growth [51,52]
This sentence does not fit here. No relation to the rest of the paragraph. Remove or complete the idea.
Response: Lines 457-459. The sentence was rewritten. “There are some other reports which are in accordance with the Gen’s et al. [49] study, indicating that bioactive cyanobacterial compounds themselves are not essential for cyanobacterial growth [51,52].”
- 992 in which P. agardhii biomass decreased or stopped growing, may suggest that dmMC-LR might play some role in the regulation of P. agardhii biomass.
Suggestion: may suggest that dmMC-LR content might be linked to P. agardhii cell density.
Response: line 505. The sentence was rewritten. “may suggest that dmMC-LR content might be linked to P. agardhii cell density” was added.
- 997
4.3. Changes in oligopeptides profiles and single compounds in P. agardhii exposed to the extracts
Suggestion : Changes in oligopeptide profiles in P. agardhii populations exposed to the extracts
Response: line 512. The subtitle was changed according to the suggestion.
Conclusions
- 1221
The obtained results showed that when lysed, P. agardhii cells release into the water compounds that may increase the biomass as well as the content of Chl-a and MCs in a natural P. agardhii population.
Response: lines 579-580. The sentence was rewritten according to the suggestion.
- 1222
MCs present in the extracts did not affect the increase in P. agardhii biomass.
Response: lines 582-583. The sentence was changed. “MCs present in the extracts did not affect the increase in P. agardhii biomass.”
- 1223
Although increasing concentrations of P. agardhii extracts did alter total MC content, changes in the relative contributions of dmMC-LR and MC-RR derivatives were found.
Response: lines 583-585. The sentence was changed according to the suggestion.
- 1226
The similarity between the oligopeptide profiles was highest for P. agardhii exposed to the each extract. As the two tested extracts differed considerably in oligopeptide composition and contained similar high concentrations of nutrients, it seems that biogenic compounds, not oligopeptides themselves, positively influenced the biomass accumulation of a natural P. agardhii population
Response: lines 586-590. The sentences were rebuilt according to the suggestion. Thank you.

Reviewer 3 Report
The manuscript is now more clear and realistic.
Author Response
Reviewer 3
The manuscript is now more clear and realistic.
Response: Thank you very much for this opinion.